# Patient-derived tumoroids from CIC::DUX4 rearranged sarcoma identify MCL1 as a therapeutic target

Willemijn Breunis[1,17], Eva Brack[2,17], Anna C. Ehlers[3,4,5,6], Ingrid Bechtold[1], Samanta Kisele[1], Jakob Wurth [1], Lieke Mous[7], Dorita Zabele [7], Fabio Steffen [1], Felina Zahnow[3], Christian Britschgi [8,16], Lorenz Bankel [8], Christian Rothermundt [9], Cornelia Vetter[10], Daniel Müller [11], Sander Botter[12], Chantal Pauli[13], Peter Bode [13], Beate Rinner[14], Jean-Pierre Bourquin [1], Jochen Roessler[2], Thomas G. P. Grünewald [3,4,5,15], Beat W. Schäfer [1] ✉, Didier Surdez [7] ✉ & Marco Wachtel [1] ✉

High-risk sarcomas, such as metastatic and relapsed Ewing and *CIC*-rearranged sarcoma, still have a poor prognosis despite intensive therapeutic regimens. Precision medicine approaches offer hope, and ex vivo drug response profiling of patient-derived tumor cells emerges as a promising tool to identify effective therapies for individual patients. Here, we establish ex vivo culture conditions to propagate Ewing sarcoma and CIC::DUX4 sarcoma as tumoroids. These models retain their original molecular and functional characteristics, including recurrent *ARID1A* mutations in CIC::DUX4 sarcoma, and serve as tumor avatars for large-scale drug testing. Screening a large drug library on a small living biobank of such tumors not only reveals distinct differences in drug response between the two entities, but also identifies a dependency of CIC::DUX4 sarcoma cells on MCL1. Mechanistically, *MCL1* is identified as a direct transcriptional target of the CIC::DUX4 fusion oncogene. Genetic and pharmacological inhibition of MCL1 induces rapid apoptosis in CIC::DUX4 sarcoma cells and inhibits tumor growth in a xenograft model. Thus, MCL1 represents a potential therapeutic target for CIC::DUX4 sarcoma. Overall, our study highlights the feasibility of drug response profiling for individual sarcoma cases and suggests that further clinical assessments of its benefit are warranted.

Sarcomas are a heterogeneous group of tumors of mesenchymal origin that are broadly subclassified into soft tissue and bone sarcomas. Ewing sarcoma (EwS) is the second most common malignant bone tumor in children and young adolescents, with an incidence of approximately one case per 1.5 million people per year. EwS can occur in any bone but is most commonly found in the diaphysis and diaphyseal-metaphyseal portions of long bones, the pelvis and ribs. Up to 20% of these tumors have an extraosseous localization[1–3]. EwS is primarily driven by FET::ETS fusion transcription factors (TFs), most

commonly EWSR1::FLI1[4–6]. In addition to classical EwS, there are rarer tumors with histological similarities, which were originally classified as Ewing-like sarcomas. Based on the identification of pathognomonic translocations, these sarcoma entities were reclassified in the 5th edition of the World Health Organization (WHO) classification of soft tissue and bone tumors as undifferentiated small round cell sarcomas of bone and soft tissue (URCS). These are further subdivided into three entities: *CIC*-rearranged sarcoma, round cell sarcoma with *EWSR1*::non-*ETS* fusions, and sarcoma with *BCOR* genetic alterations[7,8]. *CIC*-

rearranged sarcomas are the most frequent URCS entity. They are defined by *CIC*-related gene fusions, most commonly *CIC::DUX4* (*CIC::DUX* sarcoma; CDS)[7]. In the latter case, the capicua transcriptional repressor (*CIC*) gene, located on chromosome 19q13, is fused with one of two double homeobox 4 (*DUX4*) genes located on chromosome 4q35 or 10q26, respectively. The fusion event functionally alters the CIC repressor to become a strong transcriptional activator that induces an oncogenic gene expression program including target genes such as *ETV1* and *ETV4*[9]. Patients with *CIC*-rearranged sarcoma have a wide age range at presentation, with a peak incidence in young adults. The tumors are often localized in the soft tissue of the limbs or the trunk. In rare cases, osseous involvement is identified and up to 40% of patients present with metastases at diagnosis[10,11]. Currently, both EwS and URCS are treated with a similar multidisciplinary approach, including neo-adjuvant/induction chemotherapy, local treatment via surgery and/or radiotherapy, and consolidation chemotherapy. EwS tumors respond quite well to this treatment, resulting in a 3-year overall survival of 75% for patients with localized disease. In stark contrast, overall survival in patients with CDS is significantly worse, with survival rates described to be between 0% and 56%[2,10–15]. This underscores the clinical importance of the re-classification and strongly suggests the need for identifying alternative, more effective treatment approaches for CDS. While in the long term, the fusion oncogenes involved may represent the most promising therapeutic targets, the challenge of developing efficient methods to interfere with their activity as TFs has hindered the development of effective, clinically applicable inhibitors so far. Furthermore, fusion-driven tumors normally have a flat mutational landscape and the fusion protein is often the only apparent oncogenic driver present[5]. Genetic analyses aimed at identifying other driver oncogenes as therapeutic targets therefore have only limited therapeutic potential[16].

As an alternative, cells isolated from patient tumors can be directly tested with drugs to identify the most effective therapy in a personalized manner, independent of underlying molecular characteristics. Particularly for leukemia, co-clinical drug response profiling (DRP) procedures have been successfully established by different groups in recent years. Therapy guidance based on DRP data has been shown to improve outcomes in heavily pretreated patients with hematologic malignancies[17]. Hence, for blood cancer, this approach is now close to translation into clinical practice[18–20]. In contrast to leukemia, where blood sampling represents a straightforward method for obtaining tumor cells, material available from solid tumors is often limited to small needle biopsies. This limitation complicates the DRP approach, making cell expansion crucial for testing a broader range of drugs. Advances in culture conditions over the last decade have considerably improved the success rates of establishing patient-derived ex vivo models. A prominent example is the organoid model approach amenable to different types of epithelial cancers, as well as some non-epithelial tumors[21]. However, standardized culture conditions for rarer tumors such as EwS and CDS have yet to be established.

Here, we describe a DRP platform for EwS and CDS tumors. We developed culture conditions that enable the establishment of patient-derived tumoroid models from EwS and CDS tumors with high efficiency. Importantly, these models faithfully recapitulate their original tumors at both the molecular and functional drug-response levels. Differential drug sensitivity analyses reveal a strong dependency of CDS cells on the anti-apoptotic protein MCL1. This finding could pave the way for more effective therapeutic strategies for CDS.

## Results

### Tumoroid models of EwS and CDS

To expand cells derived from small biopsies of EwS and CDS tumors for patient-specific drug profiling, we used an organoid culture approach. The basis was conditions that were originally optimized for epithelial and carcinoma organoids, which are now widely used to culture organoids derived from various tissues, including sarcomas[22–27]. Tumor tissue was first dissociated mechanically (small biopsies) or enzymatically (larger biopsies), and single cells were suspended in Matrigel and plated as Matrigel domes (if <500,000 cells were available) or the cell suspension was directly plated on a thick layer of Matrigel (if > 500,000 cells were available) overlaid with culture medium (Fig. 1a). In total, 14 EwS tumors originating from 13 patients (one diagnostic-relapse pair) and four CDS tumors originating from three patients (one diagnostic-relapse pair) (see Supplementary Table 1 for clinical data) were available for this procedure. When enough dissociated cells from the patient tumor were available, a direct drug profiling was performed in parallel and used for comparison with the tumoroid approach (Fig. 1a).

In seven EwS cases (50%) and all four CDS cases (100%), we could successfully establish a tumoroid model (Fig. 1b and c). After a few days to several weeks in culture, tumor cells typically formed small clusters that further enlarged over time. After one to two passages in Matrigel domes, cells were propagated on a thick layer of Matrigel, on which they continued to grow as 3D structures (Fig. 1b). Most of the models also grew as a monolayer when plated on a thin layer of Matrigel, in some cases as 3D clusters of cells (Supplementary Fig. S1). On average, after 113 days (42-210 days) of culture, a sufficient number of cells for large-scale drug testing was available (Fig. 1c). To further optimize culture conditions, we tested whether different growth factors would accelerate tumoroid growth. We chose bFGF and EGF, both widely used in organoid protocols, and IGF1, as the IGF1R pathway plays a crucial role in EwS and other sarcoma entities. We supplemented the culture medium of three EwS and three CDS models with each growth factor and assessed cell viability after seven days of treatment. Interestingly, bFGF reduced viability of several models, in case of ES-ZH001 through rapid and strong induction of cell death as detected already after four days of treatment (Fig. 1d and Supplementary Fig. S2a-d), a finding consistent with published data on EwS cell lines[28,29]. Furthermore, in the EwS models, expression of known EWSR1::FLI1 target genes was strongly deregulated upon bFGF treatment (Supplementary Fig. S2e). In contrast, EGF and IGF1 had only minor effects (Fig. 1d). Based on these data, all tumoroid models were subsequently cultured in absence of any supplemental growth factors.

To evaluate whether the models maintained representative characteristics for the respective tumor entities, we characterized them at the molecular level. We performed DNA methylation profiling, RNA sequencing (RNA-seq) and whole exome sequencing (WES) of all the models, as well as of the original patient tumors, if material was available. We first integrated the DNA methylation profiles from the tumoroid models with a large reference cohort from the sarcoma classifier (www.molecularneuropathology.org/mnp, *n* = 1077)[30] using t-SNE dimensionality reduction. This analysis revealed close association of the tumoroids with the corresponding tumor class (Fig. 2a). Accordingly, the DNA methylation-based classification score from the sarcoma classifier revealed that with the exception of two EwS cases, samples matched the DNA methylation class of the respective tumor type (calibrated classifier scores ≥0.9 used as threshold for sarcoma subtype prediction) (Supplementary Table 2). Interestingly, the histological appearance of one of the two outlier tumors (classifier score 0.63) was non-classical for EwS, despite the presence of *EWSR1::FLI1* (see below). In contrast, classifier scores of classical EwS cell lines (data from GSE176339[31]) are mostly low (Supplementary Table 2). This suggests that DNA methylation patterns are stable during the establishment of the tumoroid models under the applied conditions, whereas in some of the available cell lines major changes in the methylation profiles have occurred.

Next, we used RNA-seq and WES data for identification of translocations and somatic mutations, respectively. A search for translocations revealed different types of *EWSR1::FLI1* fusions in the EwS samples (Fig. 2b). In addition, some of the EwS tumors also contained

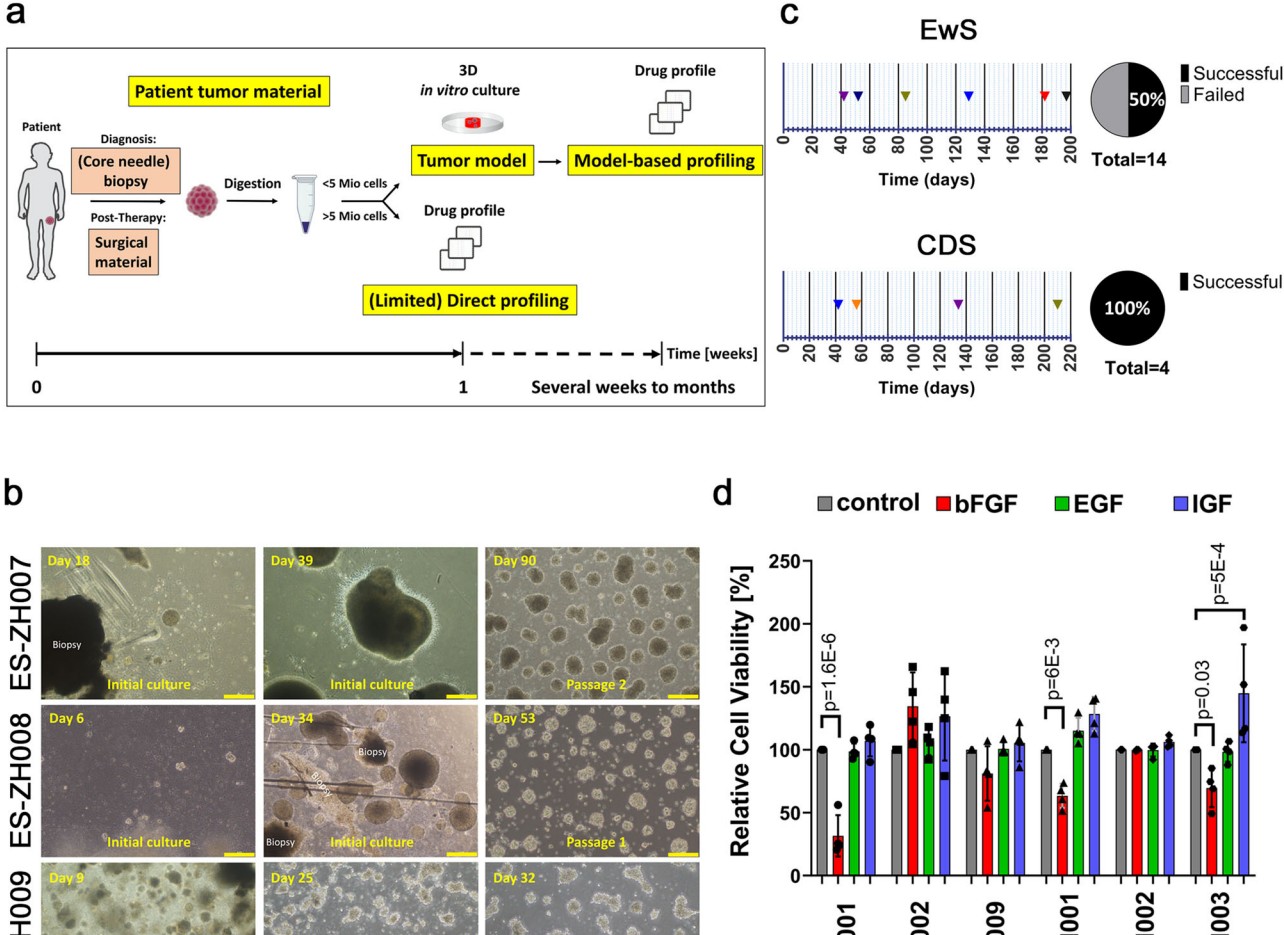

**Fig. 1 | Generation of tumoroid models from EwS and CDS tumors. a** Scheme depicting the drug profiling pipeline with separate arms for direct drug profiling (lower arm) and profiling after generation of ex vivo tumoroid models (upper arm). Created in BioRender. Wachtel, M. (2025) https://BioRender.com/t5q1jju. **b** Phase-contrast microscopy images illustrating the development of three EwS models. Images were taken at an early phase of culture in Matrigel domes (left panels) and at later time points after cell expansion in Matrigel domes or on a thick layer of Matrigel (middle and right panels, respectively). Each tumoroid was established once from human tumor material. Scale bar, 400 μm. **c** Time required to generate individual tumoroid models (left panels) and overall success rate (right panels) for EwS and CDS tumors. **d** Viability of indicated EwS and CDS cells after treatment with bFGF, EGF and IGF1 for seven days relative to cells grown in absence of growth factors. Cell viability was assessed using CTG assay. Plotted are means ± SD. $n = 4$ independent experiments, two-way ANOVA, Tukey's multiple comparisons test. Source data are provided as a Source Data file.

*STAG2* and *TP53* mutations (Fig. 2b), corroborating known mutations in EwS tumors. Additional intracellular flow cytometry revealed a loss of STAG2 expression in two out of four tested models (Supplementary Fig. S3). In case of CDS models, some commonly used fusion callers did not detect *CIC*-translocations based on RNA-seq data, similar to another case described in the literature[32], possibly due to the complex structure of the *DUX4* loci, which contain numerous pseudogenes. Archer Fusionplex analysis however confirmed the *CIC::DUX4* fusion in all CDS samples (Fig. 2c). Interestingly, two of the CDS models contained frameshift mutations in *ARID1A*, as already described for individual cases[33], suggesting that this is indeed a common mutation in CDS (Fig. 2b). Overall, our data reflects the known mutational landscape of these two types of sarcoma.

Finally, we performed a gene expression analysis with the RNA-seq dataset. We applied a comparative approach between EwS and CDS datasets to identify subtype specific gene signatures. Unsupervised hierarchical clustering with the 2,000 most differentially expressed genes revealed large differences between the two entities (Fig. 2d and Supplementary Data 1). Differential gene expression analysis with fc > 2

and fdr < 0.1 identified 2,139 and 2,544 genes up- and downregulated in CDS compared to EwS tumors, respectively (Fig. 2e and Supplementary Data 2). Among the genes upregulated in CDS were ETV4, ETV5, and WT1, all well-known markers of CDS. Gene set enrichment analysis (GSEA) using the curated Molecular Signatures Database (MSigDB) revealed the expected association of EwS and EWSR1::FLI1 target gene signatures with the EwS models (Supplementary Table 3). Interestingly, GSEA with the KEGG pathway database revealed an upregulation of different signaling pathways in CDS compared to EwS cells, including the PI3K and the Notch pathway, the former being in agreement with recently published data from CDS PDX models (Supplementary Table 3)[34]. Overall, the molecular analyses demonstrate that the specific molecular characteristics of the corresponding tumor types are maintained in our tumor models.

**Tumoroid models as tool for ex vivo drug profiling**
To investigate whether tumoroids can serve as models for therapy guidance, we performed drug response profiling (DRP) using a clinically oriented library of 245 drugs (Supplementary Table 4) applied at

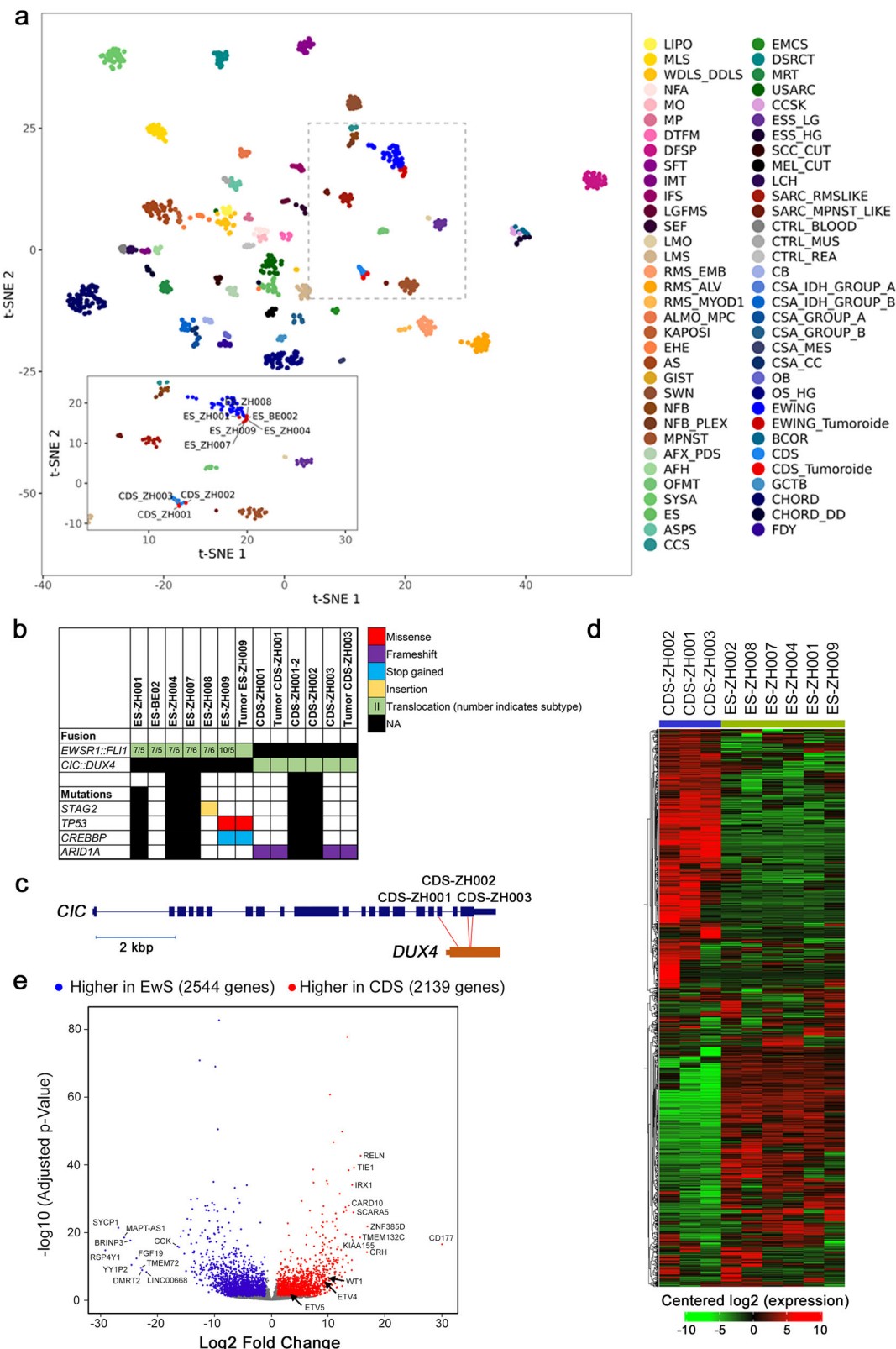

four different concentrations ranging from 10 nM to 10 μM. Cell viability was assessed by CellTiter-Glo assay after 72 h of treatment. The drug library included most of the chemotherapeutics commonly used for sarcoma treatment, as well as a range of experimental drugs targeting various signaling pathways to identify druggable dependencies. Importantly, in one CDS and one EwS case, a sufficient number of cells from the patient's tumor was available to perform upfront direct DRP.

Comparison of the response patterns from these direct screens with those from the corresponding tumoroid models revealed a high concordance (Pearson correlation coefficients r = 0.8524 and r = 0.8352, respectively) (Fig. 3a), suggesting that drug response characteristics are also maintained in our organoid models. Comparison of drug responses across different models revealed heterogeneity among samples (Fig. 3b and c). Strikingly, EwS and CDS models were clearly

**Fig. 2 | Molecular characterization of EwS and CDS tumoroid models.**
**a** Visualization of DNA methylation profiles of CDS and EwS models (n = 9) integrated with methylation profiles of the sarcoma classifier reference cohort[30] (n = 1077) using t-SNE dimensionality reduction. Samples are color-coded according to 62 tumor and three control DNA methylation classes. The inset shows a magnification of the dashed-line area and indicates the location of the individual models. For a description of abbreviations, see extended legend in the supplementary section. **b** Fusion genes and mutational landscape of EwS and CDS tumoroid models. Data from corresponding primary tumors is shown for

comparison when available. **c** Schematic representation of the *CIC* (NM_001379485.1) and *DUX4* gene architectures, with translocation breakpoints in CDS models indicated by red lines. **d** Unsupervised hierarchical clustering based on RNA-seq gene expression data from EwS and CDS tumoroid models. The 2000 most variably expressed genes were used. **e** Volcano plot depicting genes differentially expressed between EwS and CDS models. Analysis was performed with DSeq2 using fold change > 2 and FDR < 0.1. The ten most differentially expressed genes, as well as CDS markers ETV4, ETV5 and WT1, are labeled.

separated by unsupervised hierarchical clustering and PCA, with EwS cells showing greater sensitivity to many of the tested drugs than CDS cells (Fig. 3b and c). To better quantify these differences in response, we calculated a differential drug sensitivity score (dDSS)[35] for each drug in every sample. The drug sensitivity score (DSS) is computed as the normalized area under the curve (AUC) over the measured concentration range, scaling between 0 (minimal drug activity) and 100 (maximal drug activity). The dDSS score quantifies the difference between the drug response of a given sample of interest and the average drug response of a reference cohort[35]. It is calculated by subtracting the average DSS of the reference cohort from the DSS of the sample of interest. Thus, the larger the difference to the reference cohort, the more exceptional the drug response of a given sample. As a reference dataset, we used DRP data generated in our lab from 34 sarcoma tumors from different entities (23 rhabdomyosarcomas, two synovial sarcomas, and nine other sarcomas) (Fig. 3d). We first calculated a mean dDSS value for several classes of chemotherapeutics (chemoscores). These scores highlight the poorer response of CDS cells to most chemotherapeutics compared to other sarcomas, with dDSS values mostly below 0, indicating lower-than-average sensitivity (Fig. 3e). In contrast, EwS cells exhibited above-average sensitivity to most chemotherapeutic classes. Overall, these data align with the poor clinical prognosis of CDS and the relatively favorable response of EwS tumors to cytotoxic chemotherapy[10,11,13].

### Exceptional sensitivity of CDS to MCL1 inhibition
Given the urgent clinical need for better treatments for patients diagnosed with CDS, we investigated whether certain drugs might be particularly effective against them. We hypothesized that drugs eliciting exceptional ex vivo responses in a specific test case compared to a larger reference cohort, are the most promising candidates for clinical benefit in these patients. Strikingly, in four out of five CDS datasets, the drug with the highest dDSS, indicative of the highest selective effect in CDS tumors relative to the reference cohort, was the MCL1 inhibitor S63845 (Fig. 3f and Supplementary Data 3). Other drugs with high dDSS in some CDS cases included the XIAP/cIAP1 antagonist birinapant and the p300/CBP inhibitor A-485 (Supplementary Data 3), the latter aligning with the mechanism of action of CIC::DUX4[36,37]. Dose-response curves confirmed the exceptional sensitivity of CDS cells to S63845 relative to other sarcoma models, with IC50 values ranging from 1-10 nM in the most sensitive cases (Fig. 3g, and Supplementary Fig. S4 and Supplementary Table 5). Tests with BRD-810, a structurally unrelated MCL1 inhibitor, for which PRISM data from 696 cell lines are available[38], further confirmed these findings. For CDS-ZH001, CDS-ZH003, and an additional CDS model (MUG CIDUS; established at a different center), IC50 values of BRD-810 ranked 7th, 13th, and 32nd when added to the list of 696 tumor cell lines (Supplementary Fig. S5a–c). In agreement with the known high sensitivity of EwS cells to PARP inhibitors[39], talazoparib was detected among the most selective drugs for the EwS models, while for S63845 a wide range of sensitivities, including some good responses, was detected (Fig. 3f and Supplementary Fig. S4).

Based on these data, we focused on MCL1 and validated the dependency of CDS cells on MCL1 at the genetic level using a CRISPR-based knockout approach in CDS-ZH003 cells. We used two sgRNAs

directed against *MCL1*, both of which efficiently disrupted MCL1 expression (Fig. 3h), while expression levels of BCL2 and BCL-XL were not significantly affected under these conditions (Supplementary Fig. S6). MCL1 knockout was accompanied by very efficient induction of cell death, as well as depletion of the cells in a competition assay with cells transduced with a control sgRNA, measured by flow cytometry (Fig. 3h). Overall, these data demonstrate the strong dependency of CDS cells on MCL1 and suggest that this protein may serve as a therapeutic target for CDS tumors.

We next aimed to characterize these findings at the molecular level. For this, we used the MCL1 inhibitor S64315, a more clinically advanced derivative of S63845, which was included in the original library. Dose-response analysis confirmed the high sensitivity of the CDS also to this drug (Fig. 4a and b). Flow cytometric analysis of CDS-ZH003 cells stained with Annexin-V and propidium iodide after treatment with 10 and 50 nM S64315 for 24 h revealed effective and rapid induction of cell death (Fig. 4c). PARP cleavage could be detected at this early time point (Fig. 4d), as expected from an inhibitor of an anti-apoptotic protein. This was further confirmed by co-treatment of the cells with different concentrations of the pan-caspase inhibitor Z-VAD-FMK. 50 μM Z-VAD-FMK efficiently rescued cell viability after 24 h of treatment with different S64315 concentrations (Fig. 4e). Taken together, these findings demonstrate that MCL1 inhibitors quickly and efficiently induce caspase-dependent apoptosis in CDS cells.

### MCL1 is a target gene of CIC::DUX4
Next, we investigated whether the exceptional sensitivity of CDS cells to MCL1 inhibitors correlated with the expression levels of MCL1 and other anti-apoptotic BCL2 family members. We measured MCL1, BCL2 and BCL-XL levels by Western blot in three CDS and three EwS models. Both MCL1 and BCL2 were found to be highly overexpressed in CDS compared to EwS cells, whereas BCL-XL expressed levels were similar in both (Fig. 5a). Similar differences were seen at the mRNA level (Fig. 5b). To confirm these findings in a larger cohort of samples, we leveraged gene expression data from a large study of various sarcomas and normal tissues[40]. This analysis revealed that *BCL2* mRNA levels are overexpressed in CDS tumors compared to other sarcoma subtypes and to a range of normal tissues (Supplementary Fig. S7a-b), in agreement with a case study showing high BCL2 protein expression in a CDS tumor[41]. *MCL1* overexpression was less evident, but a statistically significant difference between CDS and EwS could also be seen (Supplementary Fig. S7a-b). We therefore wondered whether the high MCL1 expression levels in CDS cells result from transcriptional upregulation by the CIC::DUX4 fusion transcription factor.

To explore a potential link between CIC::DUX4 and the regulation of MCL1 and/or BCL2 expression, we analyzed published CIC::DUX4 ChIP-seq data from two different CDS cell lines for evidence of CIC::-DUX4 binding in the vicinity of *MCL1*[36]. The analysis identified two CIC::DUX4 peaks in or near *MCL1*, one in exon 2 (transcript ID NM_021960.5) and another one ~6.4 kbp downstream of the gene (Fig. 5c). High levels of H3K27Ac were detected at the same sites, suggesting that these regions may function as enhancer elements (Fig. 5c). In contrast, in case of *BCL2*, in only one of the two cell lines a small CIC::DUX4 peak was found near the last intron at the 3'-end (Supplementary Fig. S8a). We used a CRISPR interference approach to

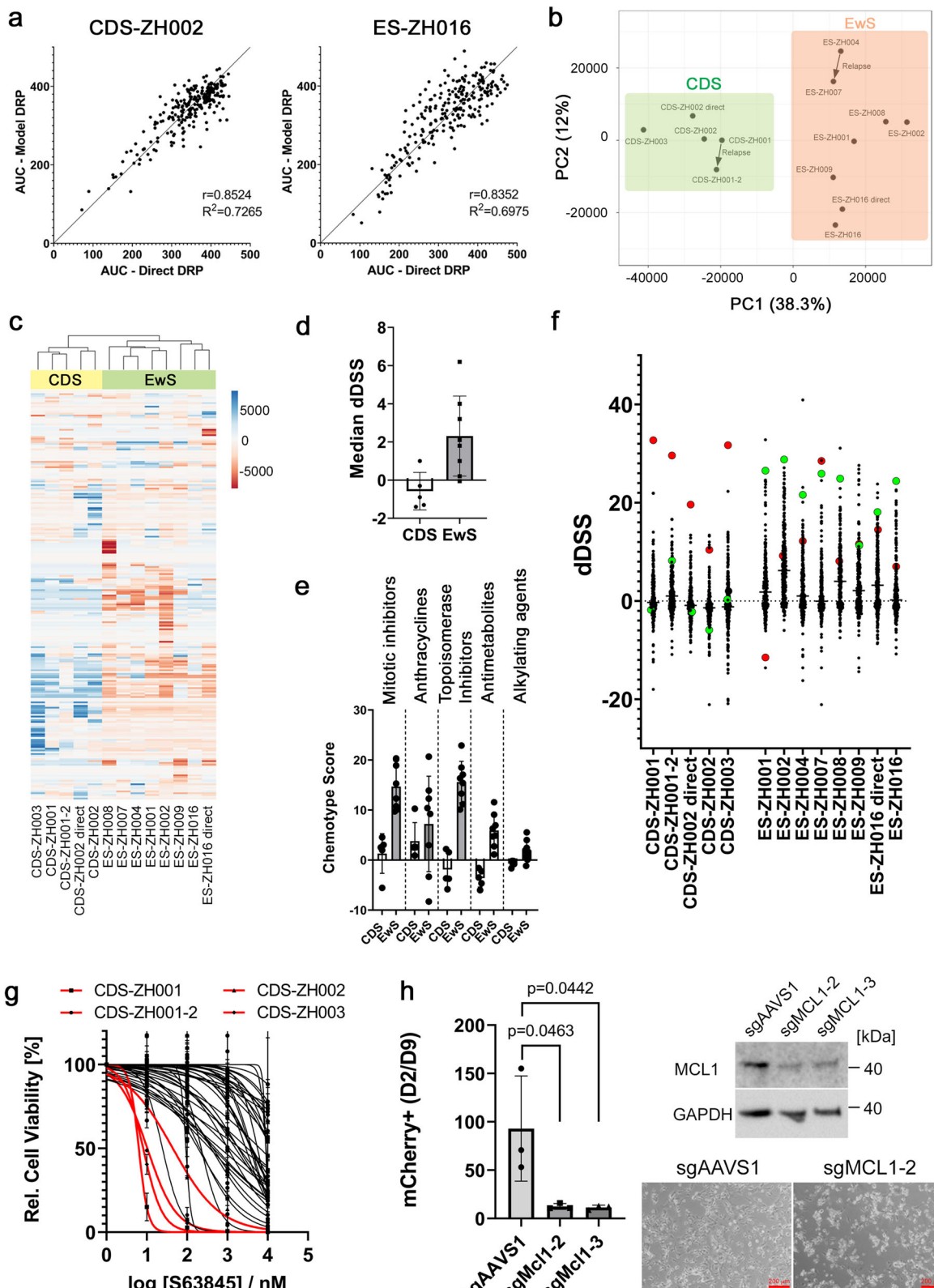

assess the relevance of these potential regulatory elements for MCL1 expression in CDS-ZH003 cells (Supplementary Fig. S8b). Targeting the downstream peak with dCas9-KRAB and specific sgRNAs significantly reduced *MCL1* mRNA levels, whereas interference with the exonic peak had no effect (Fig. 5d). This indicates that the downstream genetic element bound by CIC::DUX4 is indeed involved in the transcriptional regulation of *MCL1*.

To further validate this regulatory relationship, we silenced CIC::DUX4 expression in CDS-ZH003 cells using three doxycycline (dox)-inducible shRNAs targeting the *DUX4* portion of the fusion transcript. Dox treatment of cells stably transduced with shDUX4-1 and -3 halted cell proliferation and induced cell death within three to four days, whereas shDUX4-2 had no physiological effect (Fig. 5e and Supplementary Fig. S9a–c). To quantify silencing at the protein level,

**Fig. 3 | Comparative drug response profiling identifies MCL1 as a therapeutic target in CDS. a** Correlation of drug response of cells tested immediately after isolation from patient tumors with the corresponding tumoroid models. For correlations, area under the curve (AUC) values were calculated from dose-response curves of 245 compounds. Direct tests: $n = 1$ experiment, tumoroid models: $n = 2$ independent experiments; Pearson correlation. **b** Principal component analysis using IC50 values generated from dose-response data from 245 drugs. Arrows link diagnostic and relapse samples from the same patient. $n = 2$ independent experiments. **c** Unsupervised hierarchical clustering of indicated EwS and CDS based on IC50 values of 245 drugs, as described in **b**. **d** Median differential drug sensitivity scores (dDSS) calculated from dose-response data generated with 245 drugs. Data points indicate individual tested models ($n = 5$ CDS and $n = 8$ EwS models). Plotted are means ± SD. **e** Mean differential dDSS calculated for indicated classes of chemotherapeutics (chemotype score). Chemotype scores represent the mean dDSS of all chemotherapeutics within the indicated drug class. Data points indicate individual models ($n = 5$ CDS and $n = 8$ EwS models). Plotted are means ± SD.

**f** Individual dDSS calculated from dose-response data of 245 drugs and indicated CDS and EwS models. Data points for the S63845 and Talazoparib are labeled in red and green, respectively. **g** Dose-response curves for S63845 generated with a sarcoma reference cohort. In addition to EwS and CDS models, 23 rhabdomyosarcomas, two synovial sarcomas, and nine other sarcomas are included. Data are presented as means ± SD. CDS are highlighted in red. **h** Competition assay for evaluation of MCL1-dependency in CDS cells. *MCL1* was knocked out in CDS-ZH003 cells using CRISPR-Cas9 with sgRNAs cloned into mCherry-expressing Cas9-expression vectors. Knockout efficiency was measured by Western blot (upper right panel, representative blot from $n = 2$ independent experiments). Effects of *MCL1* knockouts on cell survival were determined by monitoring depletion of mCherry-labeled cells between day two and nine after transduction using flow cytometry (left panel). Plotted are means ± SD. $n = 3$ independent experiments, one-way ANOVA, Tukey's multiple comparisons test. Cell morphology three days after sgRNA transduction is shown in the lower right panels. Scale bar, 200 μm. Source data are provided as a Source Data file.

we compared the levels of CIC::DUX4 with those of wildtype CIC by Western Blot. Both proteins are expressed at nearly stoichiometric levels in a long and a short form, corresponding to two transcripts with different transcriptional start sites (TSSs). These forms are distinguished by a large exon near the N-terminus, as determined by CRISPR interference using sgRNAs targeting the two TSSs (Supplementary Fig. S10a-b). shDUX4-1 and -3 both reduced CIC::DUX4 levels by more than 50 percent compared to CIC, whereas shDUX4-2 had only minor effects (Fig. 5f–g). CIC::DUX4 silencing led to a marked reduction in MCL1 protein levels, most notably with shDUX4-3, as well as to a slight reduction in BCL2 levels (Fig. 5f and h). qRT-PCR detection of *MCL1* transcripts demonstrated that this effect corresponds to reduced mRNA levels (Fig. 5i). Notably, we detected a similar degree of downregulation for the well-known CIC::DUX4 target gene *ETV4* (Fig. 5i). Importantly, *GAPDH* mRNA levels were also strongly affected by CIC::DUX4 silencing and could not be used as a reference for mRNA quantification (Supplementary Fig. S11a). In contrast, *B2M* and *ANXA5* mRNA levels were both found to be CIC::DUX4 independent (Supplementary Fig. S11b), and therefore *B2M* was used as a reference gene in this experiment. Taken together, these data establish *MCL1* as a direct transcriptional target of the CIC::DUX4 fusion protein and provide a mechanistic basis for the exceptional sensitivity of CDS to MCL1 inhibition.

## MCL1 inhibition slows CDS tumor growth in vivo

Since S64315 is used in combination treatment approaches in ongoing clinical trials, we sought to identify potential drug synergies in CDS cells when combined with it. We performed a synergy screen by treating the cells with our drug library in presence or absence of 1 nM S64315 as a backbone and calculated the IC50 for each drug under both conditions. This approach revealed that S64315 exhibited the strongest synergy with inhibitors of other BCL2 family proteins, including A1331852, venetoclax, and navitoclax (Fig. 6a). Additionally, we identified several BET bromodomain inhibitors and the PI3K inhibitor PQR514 among the top hits. We further characterized the combinatorial effect between S64315 and venetoclax or PQR514 in additional CDS models and compared these effects to combinations of S64315 with chemotherapeutic agents used in the first-line treatment of CDS and EwS tumors. These chemotherapeutics are part of the so-called VIDE regimen, which includes vincristine, ifosfamide, doxorubicin, and etoposide. Bliss, ZIP, and Loewe synergy score analyses showed no synergy between S64315 and chemotherapeutic agents. In contrast, the combination of S64315 with venetoclax demonstrated clear synergism in two of the four CDS models, while its combination with PQR514 showed synergism in one model (Fig. 6b–c and Supplementary Fig. S12a–b).

Finally, we investigated the impact of MCL1 inhibition on tumor growth in vivo. We selected the CDS-ZH001 model for this purpose,

which exhibited the lowest IC50 for S64315 in ex vivo experiments. Tumor-bearing mice were treated with 20 mg/kg S64315 twice a week for three weeks. Several tumors in the treated group exhibited strong regression, and mouse survival was significantly prolonged compared to the control group (Fig. 6d). Overall, these data confirm the exceptional MCL1 dependency of CDS tumors and highlight MCL1 as a therapeutic target in these tumors.

## Discussion

Although most patients with EwS respond to standard treatment modalities, approximately one quarter remain uncured. In patients with metastatic or recurrent disease, the prognosis is markedly worse. The prognosis is even worse for CDS patients, with the majority of patients dying from the disease[10,11,13,14]. Therefore, novel therapeutic approaches to improve clinical outcomes are needed for all these patients. Personalized therapeutic strategies may help to achieve a better clinical response. Importantly, the rarity of clinically actionable driver mutations in childhood cancer in general is a major limitation in the genetic search for therapeutic targets. As an alternative, phenotypic profiling of short- to medium-term ex vivo cultures of patient-derived tumor cells has emerged as a promising tool for identifying the most effective drugs among large drug libraries[17]. The short-term approach, originally developed for blood cancers, is now increasingly used for different types of solid tumors, including sarcoma[23]. This approach requires sufficient cells from the patient tumor and is mainly suitable for tumor resections[23]. In the case of biopsies, the number of cells is a limiting factor and ex vivo expansion is often necessary, which requires appropriate culture conditions[24–26]. To this end, we developed a tumoroid culture system that enables efficient expansion of cells from EwS and CDS patient tumors. In a relevant number of cases, we were able to expand single or small clusters of cells isolated from biopsies into large tumoroids within a few weeks. At this stage, the number of available cells is sufficient to test a limited set of drugs, such as the chemotherapeutics used as first-line therapy. This timeframe may also be compatible with a co-clinical approach. Here, however, we expanded the cells for an additional one to two passages to obtain at least 10 million cells, a number required to test our drug library of 245 drugs. The average duration for this approach of 130 days is too long for a co-clinical application for therapy guidance in individual patients. However, this approach not only allowed us to test clinically approved compounds, but also to characterize the models in great detail and gain insights into unidentified relevant pathways. A beneficial byproduct of this procedure is the generation of novel models, which serve as valuable research tools for understudied tumors like CDS, facilitating future biological characterization. Still, in about half of the EwS tumors model generation failed. While the exact reason for this is unclear in individual cases, the number and viability of cells isolated from biopsy samples as well as cell-intrinsic (tumor aggressiveness) or

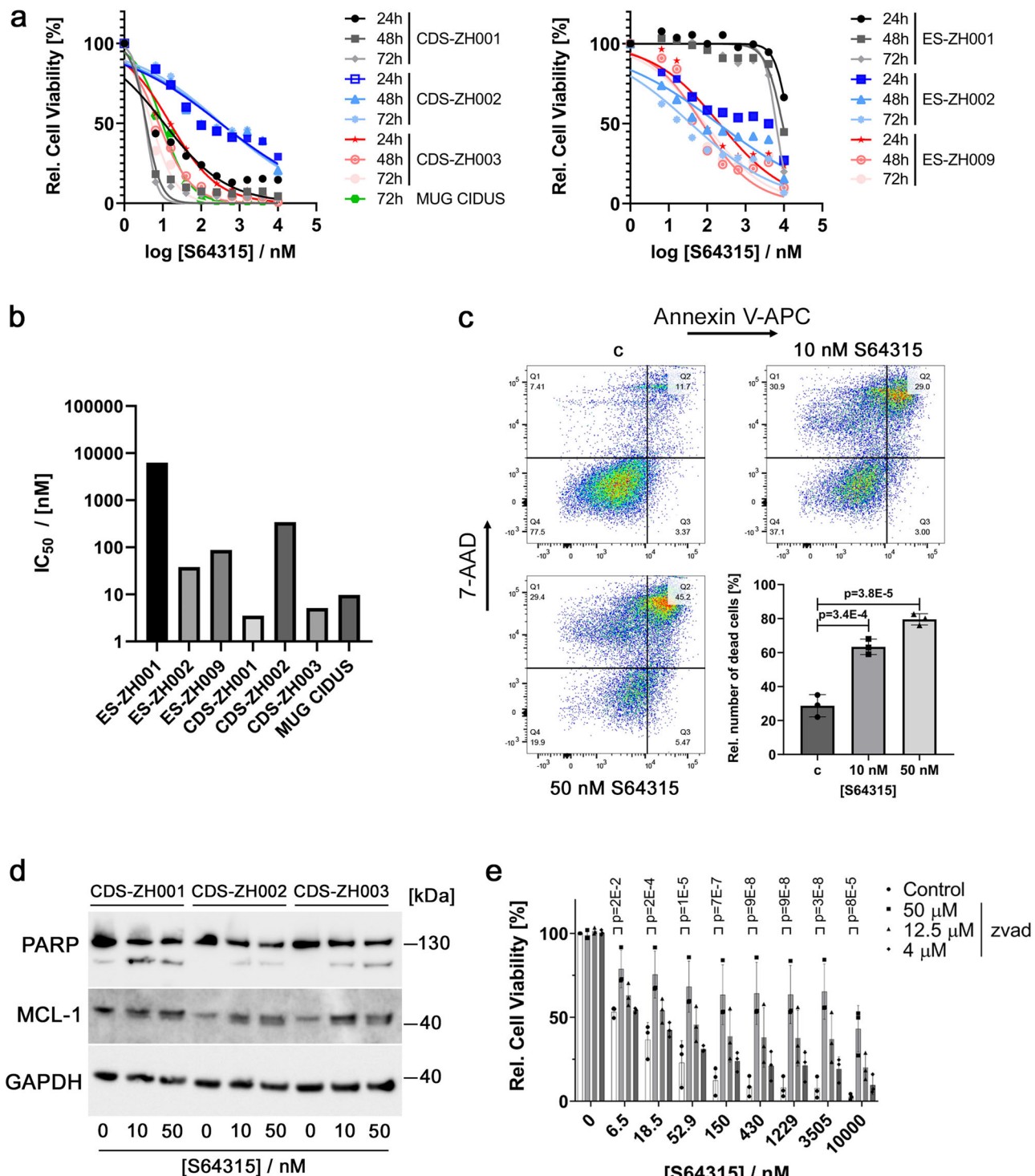

**Fig. 4 | MCL1 inhibition efficiently induces apoptosis in CDS tumor cells. a** Dose-response curves for CDS (left panel) and EwS (right panel) tumoroid models treated with the MCL1 inhibitor S64315. Cell viability was measured at the three indicated time points by CTG assay. Data are presented as means. $n = 2$ independent experiments. **b** IC50 values calculated from the data shown in a after 72 h of treatment with S64315. **c** Detection of cell death after treatment of CDS-ZH003 with the indicated doses of S64315 for 24 h using Annexin V and 7-AAD staining followed by flow cytometry. Representative pseudocolor density plots illustrate the number of live and dead cells (upper and lower-left panels). The percentage of dead cells,

defined as Annexin V and/or 7-AAD positive, is shown in the lower right panel. Plotted are means ± SD. $n = 3$ independent experiments, one-way ANOVA, Tukey's multiple comparisons test. **d** Western blot analysis of cell lysates from three indicated CDS tumoroid models after treatment with S64315 for 24 h. $n = 1$ independent experiment. **e** Viability of CDS-ZH003 cells after treatment with the indicated dose range of S64315 in presence or absence of different doses of Z-VAD-FMK. Plotted are means ± SD. $n = 3$ independent experiments, two-way ANOVA, Tukey's multiple comparisons test. Source data are provided as a Source Data file.

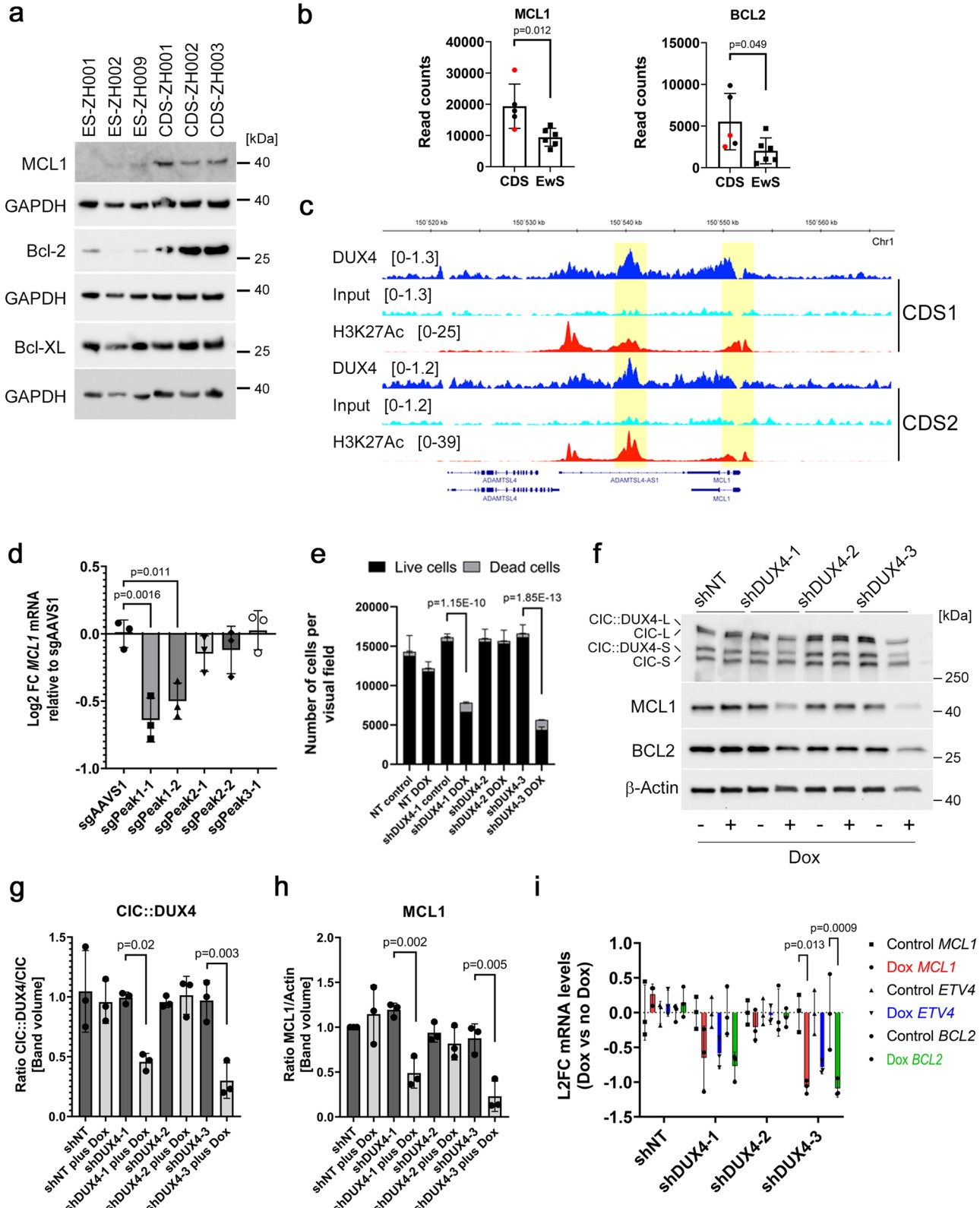

extrinsic (tumor microenvironment) factors may play an important role. Therefore, obtaining additional biopsies could further optimize the success rate and accelerate model generation.

A key finding of our study was the preservation of drug response characteristics throughout model establishment. There was no major difference between cells tested immediately after isolation and after model establishment. This is in line with published DRP data from a rhabdomyosarcoma case[42]. The models also reproduced the

difference in drug response between EwS and CDS tumors as observed in clinical practice[43]. However, the ex vivo DRP technology used here also has its limitations. It does not consider the actual tumor environment or the patient's immune system, both of which can have a major impact on treatment response. Furthermore, pharmacological differences between ex vivo cell cultures and in vivo patient physiology (pharmacokinetics, prodrug formulations and liver metabolism) introduce additional uncertainties that need to be considered.

**Fig. 5 | MCL1 is a CIC::DUX4 target gene in CDS cells. a** Western blot analysis of lysates from indicated CDS and EwS tumoroid models. *n* = 1 independent experiment. **b** mRNA levels of *MCL1* and *BCL2*, determined by normalized read count analysis of RNA-seq data. Data points indicate individual samples (*n* = 5 CDS and *n* = 6 EwS samples). Red dots indicate primary tumors, and black dots indicate tumoroid models. Data are presented as means ± SD. Unpaired two-sided t-test. **c** ChIP-seq tracks for CIC::DUX4 and H3K27Ac at the *MCL1* locus in CDS1 and CDS2 cells. CIC::DUX4 peaks further studied are highlighted in yellow. ChIP-seq data was previously published[36]. **d** Log2 fold change of *MCL1* mRNA levels in CDS-ZH003 cells after CRISPR interference with the indicated sgRNAs. Data are presented as means ± SD. One-way ANOVA, Tukey's multiple comparison test, *n* = 3 independent experiments. **e** Effect of the indicated shRNA on viability of CDS-ZH003 cells. Cells transduced with indicated doxycycline-inducible shRNA constructs were either treated with doxycycline or left untreated for five days. Live and dead cells were quantified using high content analysis. Plotted are means ± SD. Two-way ANOVA, Tukey's multiple comparison test,

*n* = 3 independent experiments. p-values were calculated for the live cell fractions. **f** Western blot analysis of lysates from CDS-ZH003 cells transduced with the indicated shRNA construct and either treated with doxycycline for four days or left untreated. The blot is representative for *n* = 3 independent experiments. **g** Quantification of CIC::DUX4 silencing. Western blot bands of the short forms of CIC::DUX4 and CIC were quantified by densitometry, and the CIC::DUX4-to-CIC ratio was calculated. Plotted are means ± SD. Two-way ANOVA, Tukey's multiple comparison test, *n* = 3 independent experiments. **h** Quantification of MCL1 protein downregulation. MCL1 and β-Actin Western blot bands were quantified by densitometry and the ratio was calculated. Plotted are means ± SD. Two-way ANOVA, Tukey's multiple comparison test, *n* = 3 independent experiments. **i** Log2 fold change of mRNA levels of *MCL1*, *BCL2* and *ETV4* in CDS-ZH003 cells after silencing of *CIC::DUX4* with the indicated shRNA. *B2M* was used as reference gene. Plotted are means ± SD. Two-way ANOVA, Tukey's multiple comparison test, *n* = 3 independent experiments. Source data are provided as a Source Data file.

Prospective co-clinical studies are needed to test whether ex vivo response is predictive of in vivo response in patients. Encouragingly, in a recent prospective observational study of the DRP approach with a small set of different pediatric solid tumors, DRP-guided therapy led to significantly improved treatment responses compared to non-guided therapy[44], mirroring findings previously reported in leukemia[17].

Based on our differential drug sensitivity analysis, we identified an exceptional dependency of CDS cells on MCL1, suggesting that this protein could serve as a therapeutic target. While this dependency has not been described yet, other therapeutic vulnerabilities of CDS cells have been reported recently. This includes the sensitivity of CDS cells to AKT/mTOR-inhibitors and trabectedin[34]. The combination of the PI3K/mTOR inhibitor dactolisib (NVP-BEZ235) with trabectedin has been shown to be effective not only against local tumor growth, but also against metastatic spread[34]. We did not observe exceptional sensitivity to dactolisib either as single agent or in combination with S64315 in our CDS models; however, we detected a synergy between the PI3K inhibitor PQR514 and S64315 in one of the models. Furthermore, our comparative gene expression analysis also hinted towards an activation of the PI3K pathway in CDS cells. Other studies demonstrated that p300 may be a promising candidate target for CDS treatment, which is in line with the mechanism of action of CIC::DUX4 as p300 recruiter[36,37]. In line with these studies, we observed a high sensitivity of one of the CDS models against A-485.

The IC50 values of the MCL1 inhibitor S63845 in CDS were comparable to those found in hematological malignancies, which are currently the focus of MCL1-targeted therapy, and significantly lower than those observed in almost all tested solid tumor cell lines[45]. Several clinical studies testing the effect of different MCL1 inhibitors have been started a few years ago with some of them still ongoing[46]. Importantly, cardiac problems have been revealed as a major dose-limiting side effect in both a study with S64315 in AML/MDS (ClinicalTrials.gov ID NCT02979366), as well as in a study with AMG397 in different hematological malignancies[47]. The exact mechanism of MCL1 inhibitor mediated cardiotoxicity is not clear and reasons are probably multifactorial. However, conditional knockout studies of MCL1 in the mouse detected an important role for MCL1 in cardiomyocyte survival[48]. Therefore, the observed clinical side effects are likely to reflect this function and potentially limit the therapeutic window of MCL1 inhibitors. The MCL1 dependency of other cell types, including haematopoietic stem cells and lymphocytes, might lead to additional limitations for clinical use[49,50]. Combining MCL1 inhibitors with agents targeting other BCL2 family members may enable dose reduction and mitigate these issues. Indeed, in our drug combination screen we detected a strong synergy between MCL1 and BCL2 inhibitors. Further clinical development of such a combination appears feasible, since in a recent phase 1 study combining S64315 and venetoclax (NCT03672695) no new toxicity signals were detected, even though the trial was stopped due to lack of clinical efficacy. New MCL1 inhibitors, like BRD-

810, which has a short half-life and thereby undergoes rapid systemic clearance, are currently in development and may also mitigate cardiac side effects[38]. Alternatively, the development of a degrader variant of one of the available MCL1 inhibitors, analogous to the PROTAC variant of the BCL2 and BCL-XL dual inhibitor navitoclax, may allow for a further reduction in drug dose and help avoid on-target toxicities[51].

Overall, our study of patient-derived tumor models identified MCL1 as therapeutic target in CDS. Moreover, as a proof of concept, this study highlights the feasibility of DRP for individual sarcoma patients. Implementation of the pipeline into clinical studies is highly warranted to evaluate the correlation with clinical response in prospective manner.

## Methods

### Ethical considerations
Patients and/or their legal guardians provided written informed consent prior to trial entry, allowing the tumor material or blood to be biobanked for further research and for the publication of their images. Use of the patient material for the experiments was approved by the ethics committee of the Kanton Zurich (BASEC number 2020-01609).

### Patient samples
Patients included both male and female individuals. All Patients, irrespective of age, that were diagnosed in one of the participating centers with a pathology-confirmed EwS or CDS and from which enough fresh biopsy material was available, were eligible. This included the primary diagnosis as well as progression or relapse.

### Sample collection process
Fresh biopsy material was either used directly or frozen in either CryoStor10 freezing medium (STEMCELL Technologies, 7930) or DMEM (Sigma-Aldrich, D5671) supplemented with 10% FBS and 10% DMSO at −80 °C in a cell freezing container at the participating sites. The frozen tumors were then shipped to the laboratory in Zurich where the tumoroid cultures were established.

### Tumoroid culture
Fresh or frozen biopsies from patient tumors were first washed with HBSS buffer (Sigma-Aldrich, H6648) and then cut into smaller fragments using two scalpels. For very small biopsies, the fragments were directly mixed with Matrigel and cultured as described below. Otherwise, the tumor material was transferred to a microcentrifuge tube and digested with 1 ml of digestion buffer containing 0.125 mg/ml Liberase DH (Roche, 5401054001) and 1 mM $MgCl_2$ in HBSS buffer for 2–4 h, until the tissue was completely dissolved. The digestion progress was monitored by counting cells in the supernatant at several time points. Cells were then pelleted by centrifugation for 2 min at 350 g. In case of the presence of red blood cells, these were lysed by incubation in 1 ml

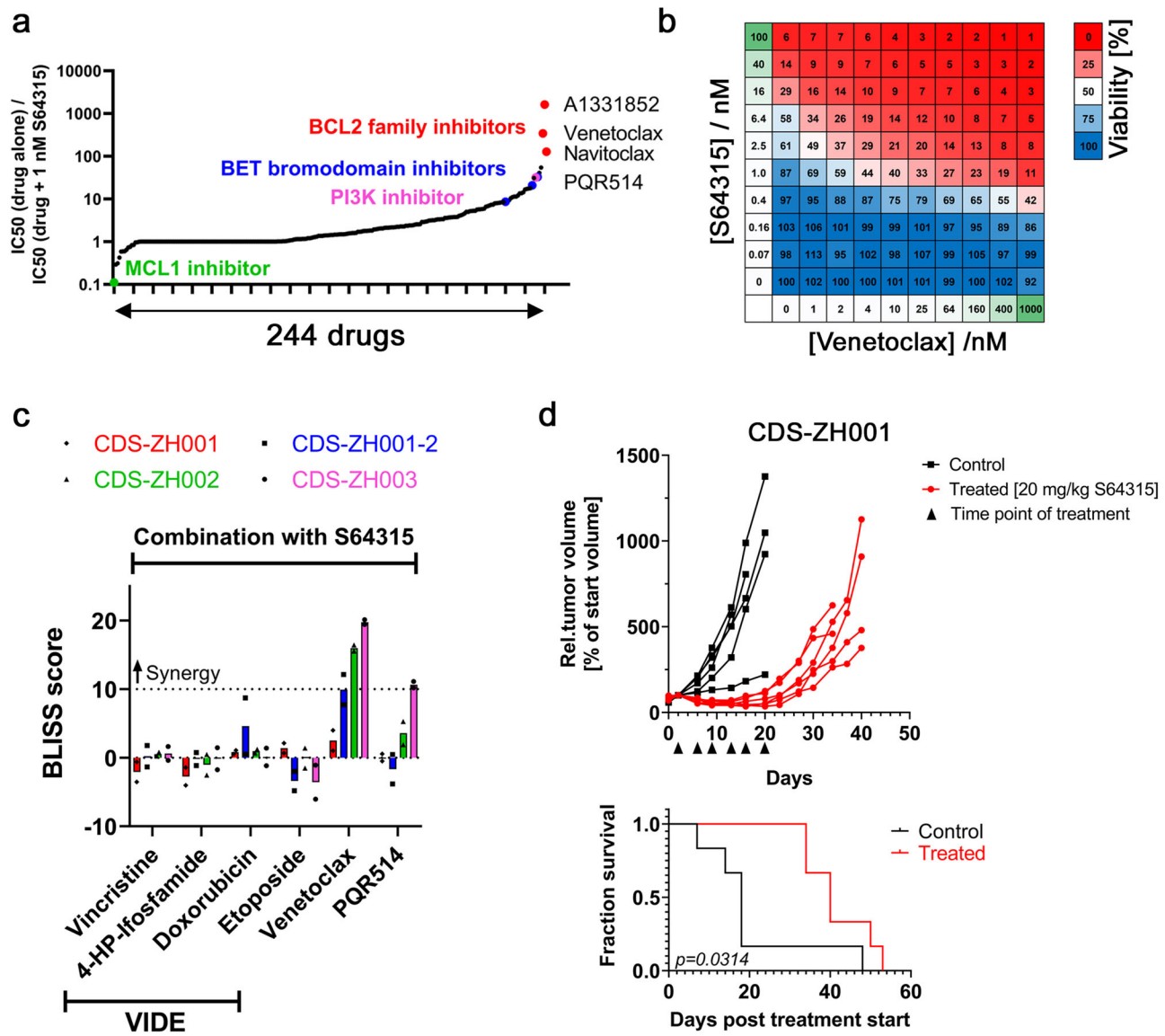

**Fig. 6 | Inhibition of MCL1 slows growth of CDS xenograft tumors in vivo.**
**a** Ratio of IC50 values for 244 drugs determined with CDS-ZH003 cells in presence or absence of 1 nM S64315. Drug classes showing high cooperativity with S64315 are highlighted in red (BCL2-family inhibitors), blue (bromodomain inhibitors), and pink (PI3K inhibitors). *n* = 2 independent experiments. **b** Combination matrix used to test the synergy between the MCL1 inhibitor S64315 and the BCL2 inhibitor venetoclax. The relative number of viable CDS-ZH003 cells as determined by image-based analysis after 72 h of treatment is depicted. *n* = 2 independent experiments. **c** Bliss synergy scores for the indicated CDS models after treatment with S64315 in combination with different chemotherapeutics, venetoclax or PQR514 for 72 h. Calculation was based on the number of viable cells, determined by image-based analysis. *n* = 2 independent experiments. **d** Growth curves of xenograft tumors in individual female mice transplanted with CDS-ZH001 cells and treated twice a week with either vehicle (black) or 20 mg/kg S64315 (red) (upper panel). Black triangles indicate the time points of drug application. Survival curves calculated with the data from the upper panel (lower panel). Log-rank (Mantel-Cox) test. Source data are provided as a Source Data file.

red blood cell lysis buffer for 10 min in the dark. Cells were then pelleted as before and washed once with HBSS buffer. The resulting cell pellet was suspended in 100–150 µl Matrigel (Corning, 354234) and the cell suspension was used to form 3D Matrigel domes in the center of a 6-well. After solidifying of the Matrigel, the domes were overlaid with 3 ml Adv.DMEM/F12 medium (Thermo Fisher Scientific, 12634028) supplemented with 100U/ml Penicillin/Streptomycin (Thermo Fisher Scientific, 15140122), 2 mM Glutamax (Thermo Fisher Scientific, 35050-061), 1xB27 (Thermo Fisher Scientific, 17504044), 1.25 mM N-acetylcysteine (Sigma-Aldrich, A9165), 5 µM A83-01 (Tocris Bioscience, 2939) and Matrigel (1:1000). The cells were then cultured at 37 °C and 5% $CO_2$ and the medium was exchanged three times a week. To prevent anoikis after tissue dissociation, 10 µM ROCK inhibitor Y-27632 (Focus Biomolecules, 10-2301) was added to the medium

for the first two to three days of culture. The cells were passaged when larger tumoroids became visible. For this, the Matrigel domes were washed from the plate in 1 ml Cultrex organoid harvesting solution (R&D Systems, 3700-100-01) and incubated on ice in a microcentrifuge tube until the Matrigel was completely dissolved. After pelleting, the organoids were dissociated with Accutase (Sigma-Aldrich, A6964) diluted 1:1 with PBS for 3-5 min at 37 °C. After washing once with PBS, cells were either filtered through a 70 µm cell strainer and used for the drug profiling or directly plated again either in Matrigel domes in 6 well plates (early passages) or on a thick layer of Matrigel in cell culture flasks (later passages). For passaging of tumoroids growing on a thick layer of Matrigel, the Matrigel was dissolved by incubation with Cultrex organoid harvesting solution at 4 °C under constant shaking for 30-60 min.

For testing the effect of growth factors on tumoroid growth, the culture medium was supplemented with 20 ng/ml bFGF (Peprotech, #100-18B), EGF (Peprotech, AF100-15) or IGF-I LR3 (Peprotech, #100-11R3).

## Genomic analyses

The genomic data evaluated in this study was produced and kindly provided by the INFORM program[16,52–54].

## RNA-seq

The libraries of the extracted RNA samples were prepared using the TruSeq Stranded RNA Library Prep kit (Illumina). The libraries were sequenced on one NovaSeq 6 K PE 100 S1 flow cell as 2x650M reads per lane by the NGS Core Facility of DKFZ (Heidelberg, Germany).

The analysis of the sequencing reads was then performed in Galaxy (https://usegalaxy.eu/) with the following tools. Illumina adapters were trimmed off using Cutadapt and quality control was performed with FastQC. RNA-seq reads were then aligned to the GRCh38 reference genome using RNA STAR. Read counts per gene were determined using featureCounts and normalization between samples was performed with DESeq2. Fusion genes were detected using Arriba. Relative gene expression analysis was performed with iDEP1.1[55].

## Whole exome sequencing

The libraries of the extracted DNA samples were prepared using the Sure Select XT HS Agilent + Human All Exon V7 protocol by the NGS Core Facility of DKFZ. The libraries were sequenced on one NovaSeq 6k PE 100 S2 flow cell as 1 × 1650 M reads per lane by the NGS Core Facility of DKFZ (Heidelberg, Germany). Analysis of the sequencing reads was then performed in Galaxy (https://usegalaxy.eu/) with the following tools. Trimming and filtering of reads was done with Trimmomatic. The sequencing reads were then aligned to the hg19 reference genome using BWA-MEM. Reads were filtered with the Filter BAM tool using the following parameters: mapquality ≥1, ismapped =yes, ismatemapped =yes. Duplicate reads were removed with RmDup. Positional distribution of insertions and deletions in the input was homogenized by using left realignment with BamLeftAlign with maximum number of iterations=5. Read mapping quality was recalibrated with CalMD with a coefficient to cap mapping quality of poorly mapped read=50. Reads were refiltered based on mapping quality using Filter BAM datasets with mapquality ≤254. Somatic variants were identified using VarScan somatic with a minimum base quality=28 and a minimum mapping quality=1. Variants were annotated with SnpEf eff.

## Methylation profiling

The extracted DNA samples were prepared and analyzed by Infinium MethylationEPIC BeadChip Array by the Microarray Core Facility of the DKFZ.

Raw methylation array data from 1077 sarcoma samples spanning 65 methylation classes was obtained from the public repository Gene Expression Omnibus (GSE140686) and processed alongside methylation data from 9 of our patient-derived tumoroid models as previously described[30]. For dimensionality reduction analysis, the 10,000 most variable CpG probes based on their standard deviation across all samples were selected. These batch corrected and normalized methylation beta values were then used to embed the high-dimensional methylation patterns using t-distributed Stochastic Neighbor Embedding (t-SNE), as implemented in the Rtsne R package (v0.17), with 3000 iterations and perplexity set at 30.

## Identification of *CIC::DUX4* translocation breakpoints

*CIC::DUX4* fusions were detected with the Archer FusionPlex assay using the sarcoma v2 panel.

## Preparation of the drug library for the drug response profiling

The 245 drugs of the library were purchased either pre-dissolved in DMSO as a 10 mM solution or as powder (Selleckchem). 5 mM and 0.5 mM stock solutions were then prepared in DMSO (238 drugs) or water (7 drugs) and transferred to 384-well source plates (Labcyte; PPL-0200) qualified for an Echo dispenser (Labcyte). The Echo dispenser was used to dispense the drugs into 384-well cell culture plates (Greiner, 7.781 098). Four wells were used per drug and filled with either 5 nl of drug solution from the 0.5 mM source plate or 5 nl, 50 nl, or 500 nl of drug solution from the 5 mM source plate, corresponding to final drug concentrations of 10 nM, 100 nM, 1,000 nM, and 10,000 nM, respectively. DMSO volumes were normalized using the Echo dispenser. The pre-drugged plates were sealed with an adhesive aluminum cover (Sigma-Aldrich, Z721530) and stored at -80 °C for up to several months until use. On the day of use, the plates were equilibrated to room temperature, centrifuged, and filled with 25 µl pre-warmed culture medium per well using a Mantis liquid handler (Formulatrix). The drug plates were then incubated at room temperature overnight under constant shaking on an orbital shaker protected from light. The next day, 2 µl of each drug solution was transferred to the plate containing the cells in 18 µl medium using a 24-channel pipette. Screening parameters are summarized in Supplementary Table 6.

## Drug response profiling

For drug response profiling, 10,000–20,000 cells in a volume of 20 µl were plated per 384-well coated with Matrigel. For the complete library, a total of 1100 wells were used. The next day, the medium was changed and the drugs were added to the cells and incubated for a total of 72 h. Cell viability was then determined with the CellTiter Glo 3D assay (Promega, G9682) according to the instructions of the manufacturer. To assess cell death, cells were stained with 10 µM Hoechst 33342 and 1 µg/ml propidium iodide for 15 min, followed by image-based analysis with an Operetta CLS high-content analysis system. Live and dead cells were quantified using the Harmony software.

AUC and IC50 values were calculated by nonlinear regression using GraphPad Prism 8.0.0. dDSS values were calculated using published procedures[35].

## Flow cytometry

For analysis of cell death, cells were grown and treated in 6-well plates. After treatment, remaining adherent cells were detached with Accutase (diluted 1:2 with PBS) (Sigma-Aldrich, A6964), and combined with floating cells in a centrifuge tube. After washing twice with PBS, single cells were resuspended in 100 µl Annexin V binding buffer (BD Biosciences, #556454) and stained with 5 µl Annexin V-APC (BD Biosciences, #550474) and 5 µl 7-AAD (BD Biosciences #559925) for 15 min at room temperature in the dark. After dilution with an additional 400 µl binding buffer, cells were filtered through a cell strainer into a FACS tube and analyzed on a Fortessa flow cytometer (BD Biosciences).

For STAG2-intracellular staining, cells were grown in T25 flasks and detached using Accutase (Sigma-Aldrich, A6964), 1:2 diluted with PBS. After washing twice with PBS, cells were fixed in PBS/1% PFA (Thermo Fischer Scientific, 28906)/0.5% BSA (Sigma-Aldrich, A2153) for 15 minutes on ice. Cells were washed twice with PBS/0.5% BSA and stored in PBS/0.5% BSA at 4 °C for up to one week before staining. Cells were permeabilized with PBS/0.5% Tween-20 (Sigma-Aldrich, P9416)/0.5% BSA (permeabilization buffer), and blocked in PBS/0.5%Tween-20/3% BSA for 1 h at RT. Cells were stained with a primary antibody against STAG2 (Santa Cruz, sc-81852) diluted 1:500 in permeabilization buffer, for 2 h at RT. Cells were washed twice with permeabilization buffer and incubated with Alexa Fluor™ 647-conjugated secondary antibody (Thermo Fischer Scientific, A-21235) diluted 1:400 in permeabilization buffer, for 1 h at RT. After two washes with

permeabilization buffer, cells were resuspended in PBS/0.5% BSA and analyzed on a Fortessa flow cytometer (BD Biosciences).

## Western Blot

Cell lysates for Western blots were generated using RIPA buffer (Thermo Fisher Scientific, #89900), supplemented with Complete Mini Protease Inhibitor cocktail (Sigma-Aldrich, #11697498001). Proteins were separated using NuPAGE™ Novex™ 4–12% Bis-Tris gels (Thermo Fisher Scientific) and transferred to nitrocellulose membranes (GE Healthcare Life Sciences) by wet blotting. Membranes were then blocked with 5% milk in TBS/0.05% Tween-20 for 20 min, followed by incubation with the primary antibody diluted at 1:1000 in blocking buffer overnight at 4 °C. After three washing steps with TBS/0.05% Tween-20 for 5 min, membranes were incubated with a horseradish peroxidase-linked secondary antibody (Cell Signaling, #7074) diluted 1:2000 in blocking buffer for 1 h at RT. After three additional washing steps with TBS/0.05% Tween-20 for 5 min and one final wash step with TBS for 1 min, proteins were detected by chemiluminescence using either the Pierce™ ECL or the SuperSignal Western blotting reagent (both from Thermo Fisher Scientific) and a ChemiDoc™ MP imaging system (BioRad). Antibodies used included the following ones from Cell Signaling: anti-MCL1 (#4572S), anti-BCL-XL (#2764S), anti-PARP (#9542S), anti-β-Actin (#4970) and anti-GAPDH (#2118). Antibody against BCL2 was purchased from Abcam (ab32124) and against CIC from Novus Biologicals (#NB110-59905SS).

## CRISPR-mediated gene knockout

Gene-specific sgRNAs were cloned into LentiCRISPR-mCherry vectors (Addgene #75161). The following sgRNAs were used: sgAAVS1 (negative control; GGGGCCACTAGGGACAGGAT), sgMCL1-2 (CGAGTTG-TACCGGCAGTCGC) and sgMCL1-3 (CTGGAGACCTTACGACGGGT). sgRNA constructs were transfected into Lenti-X™ cells (Takara Bio) together with pMD2.G (Addgene #12259) and psPAX2 (Addgene #12260) packaging vectors. The next day, the medium was changed. Sixty hours after transfection, supernatants containing lentiviral particles were harvested and concentrated using Amicon Ultra columns (Merck #UFC910008) with a cut-off of 100 kDa. For transduction of CDS target cells, viral concentrates were diluted 1:10–1:20 in medium and added to the target cells grown in 24- or 6-well plates. Infection rate was enhanced by spinoculation with 1500 g for 90 min at 32 °C. After overnight incubation, the medium was changed.

## Competition assay

CDS cells were transduced with mCherry-expressing LentiCRISPR sgRNA constructs. Three days after transduction, transduced cells were mixed with untransduced cells as a reference and the fraction of mCherry-positive to mCherry-negative cells was determined by flow cytometry on a Fortessa flow cytometer (BD Biosciences). Seven days later, the fraction was again determined, and the ratio of mCherry-positive fractions between day 2 and day 7 was calculated.

## Quantitative PCR

Total RNA was isolated from cultured cells with the RNeasy Plus Mini Kit (Qiagen). cDNA from 1 µg total RNA was generated using the High Capacity cDNA Reverse Transcription Kit (Thermo Fisher Scientific). mRNA levels of genes of interest were then determined with specific TaqMan™ assays including assays for *MCL1* (Hs06626047_g1), *BCL2* (Hs04986394_s1), *ETV4* (Hs00383361_g1), *GAPDH* (Hs02758991_g1) (all from Thermo Fisher Scientific), *B2M* (Hs.PT.58 v.18759587) and *ANXA5* (Hs.PT.56a.38876508) (both from Integrated DNA Technologies) and the TaqMan™ Gene Expression Master Mix (Thermo Fisher Scientific). Relative gene expression levels were calculated using the ΔΔCT method with *GAPDH*, *B2M* or *ANXA5* as reference gene.

## CRISPR interference

For CRISPR interference, dCas9-KRAB was cloned into a lentiviral expression plasmid (Addgene #134966), replacing wildtype Cas9. The plasmid also contains GFP as a selection marker. CDS-ZH003 cells were transduced with the construct and sorted for uniform GFP expression. The selected cells were then transduced with different sgRNA expression constructs (sg_shuttle_RFP657, Addgene #134968). For interference with the CIC::DUX4 binding sites near *MCL1*, the following sgRNAs were used: sgPeak1-1 (GAGTCATAACCAGCCCAGTA), sgPeak1-2 (CACTGGCAGTA GACTTATAC) (both directed against the MCL1 downstream enhancer) and sgPeak2-1 (TTCAGTTGAATACTCTTCAG), sgPeak2-2 (CCATGAA AAAATAAGTCACC), and sgPeak3-1 (GAACAACAGTCTTAGATGAT) (all directed against the genic CIC::DUX4 binding site). Two to three days after transduction, RNA was isolated and *MCL1* mRNA levels were measured by qRT-PCR.

To evaluate *CIC* transcriptional start sites active in CDS-ZH003 cells, the following sgRNAs were used: sgCIC-1 (TGGCAGCGG-TAGCGGCACGA), sgCIC-2 (AATCGAGAGGGAGAGCCGGA), and sgCIC-3 (CCTGCCTCCCGCCGCCCGGG). Two to three days after transduction, protein lysates were generated and CIC::DUX4 and CIC protein levels were evaluated by Western blot.

## shRNA-mediated CIC::DUX4 silencing

Three different shRNAs targeting sequences in the C-terminal region of *DUX4* were tested for silencing of CIC::DUX4. The following shRNAs were used (21-nt targeting sequence): shDUX4-1 (AGGCG-CAACCTCTCCTAGAAA), shDUX4-2 (GGCTCTGCTGGAGGAGCTTTA) and shDUX4-3 (CAACCTCTCCTAGAAACGGAG). A non-targeting sequence was used as negative control (CAACAAGATGAA-GAGCACCAA). The shRNA sequences were cloned into the pRSIT12 vector under a doxycycline-inducible promoter (Cellecta). CDS-ZH003 cells were transduced with the constructs and shRNA expression was induced by treating the cells with 1 µg/ml doxycycline. The culture medium was changed every other day.

## ChIP-seq analysis

CIC::DUX4 binding sites in the vicinity of *MCL1* were analyzed in the Integrative Genomics Viewer (IGV) using published ChIP-seq data downloaded from the Gene Expression Omnibus (GEO) database under the accession number GSE248117[36].

## In vivo tumor treatment

NSG mice (NOD.Cg-Prkdc< scid> Il2rg<tm1Wjl >/SzJ; Charles River F, L'Arbresle, France) were bred in-house. Health screens were conducted in accordance with FELASA guidelines to confirm their pathogen-free status once per quartal. Animals were housed in groups of 4-6 mice per individually ventilated cage in 12 h light/dark cycle, with controlled room temperature (21 °C) and relative humidity (40-60%). The in vivo study exclusively examined female. To establish xenografts, 5 million CDS-ZH001 cells were injected s.c. into the flank of 8-10 weeks old mice. Tumor-bearing mice were randomized into treatment and control cohorts of 6 animals when the average tumor size reached ~100 mm³. S64315 (MIK665; Medchemexpress) was dissolved in 20% 2-Hydroxypropyl-beta-cyclodextrin (Sigma-Aldrich), 25 mM HCl[56]. Drug and vehicle were administered by i.v. injection twice a week. Tumor size was measured three times a week with a caliper, and mouse weight was measured twice a week. No mice had to be euthanized due to severe weight loss ( > 15% of baseline). All mice were euthanized when tumor volumes exceeded 1000 mm³. All animal experiments were approved by the veterinary service of the canton of Zurich and were performed according to the animal license ZH013/2021.

**Reporting summary**

Further information on research design is available in the Nature Portfolio Reporting Summary linked to this article.

## Data availability

WES, RNA-seq and DNA methylation data from tumoroid models is deposited in the European Genome-Phenome Archive under the study ID EGAS00001008039. Due to patient data protection, the data are available upon request to the data access committee (DAC). The links to the DAC can be found under the dataset IDs EGAD00001015608 (WES and RNA-seq; https://ega-archive.org/datasets/EGAD00001015608) and EGAD00010002760 (DNA methylation data; https://ega-archive.org/datasets/EGAD00010002760). The data will be made accessible to all interested researchers, with no limitations regarding the time or purpose of data use. Previously published ChIP-seq data[36] was downloaded from the Gene Expression Omnibus (GEO) database under the accession number GSE248117. The remaining data are available within the article, supplementary information or source data file. Source data are provided with this paper.

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

## Acknowledgements

We thank the NGS Core Facility at the German Cancer Research Center (DKFZ) for providing excellent RNA-seq, WES and methylation profiling services. We would also like to express our sincere thanks to Carsten Maus and Erjia Wang (Genomics and Proteomics Core Facility, DKFZ), Lena Weiser and Gregor Warsow (Omics IT and Data Management Core Facility, DKFZ) for their highly dedicated support in data management and processing and Robert Autry, Gnanaprakash Balasubramanian, Christopher Previti, and Rolf Kabbe (Division of Pediatric Neurooncology, DKFZ) for their sincere and dedicated contribution to the bioinformatics analyses. Finally, we thank Rishi Puram and Todd Golub (Broad Institute of MIT and Harvard) for providing BRD-810. The work was supported by grants from the Swiss National Science Foundation (3100-175558 to BWS and 10000473 to DS), the Cancer League Switzerland (KLS-5143-08-2020 to BWS and KFS-5422-08-2021-R to DS), the Childhood Research Foundation Switzerland (to BWS), the Berner Stiftung für krebskranke Kinder und Jugendliche (to JR), the European Union's Horizon 2020 research and innovation program under the Marie Skłodowska–Curie grant agreement No. 956285 (VAGABOND) (to BWS), and a clinical research focus grant from the University of Zurich (to BWS, DS, and JPB). The laboratory of TGPG is supported by the Barbara and Wilfried Mohr Foundation and co-funded by the European Union (ERC, CANCER-HARAKIRI, 101122595). Views and opinions expressed are, however, those of the authors only and do not necessarily reflect those of the European Union or the European Research Council. Neither the European Union nor the granting authority can be held responsible for them. ACE was supported by a scholarship from the German Cancer Aid and the German Academic Scholarship Foundation. The INFORM program is financially supported by the German Cancer Research Center (DKFZ), several German health insurance companies, the German Cancer Consortium (DKTK), the German Federal Ministry of Education and Research (BMBF), the German Federal Ministry of Health (BMG), the Ministry of Science, Research and the Arts of the State of Baden-Württemberg (MWK BW); the German Cancer Aid (DKH), the German Childhood Cancer Foundation (DKS), RTL television, the aid organization BILD hilft e.V. (Ein Herz für Kinder) and the generous private donation of the Scheu family.

## Author contributions

Conceptualization: W.B., E.B., B.W.S., T.G.P.G., D.S., M.W. Methodology: M.W. Formal analysis: A.E., M.W. Investigation: F.Z., I.B., S.K., J.W., L.M., D.Z., M.W. Resources: W.B., E.B., J.R., C.B., L.B., C.R., D.M., J.P.B., F.S., S.B., C.P., P.B., B.R., C.V. Writing-original draft: W.B., E.B., I.B., T.G., M.W. Writing-review and editing: W.B., E.B., D.S., T.G.P.G., B.W.S., M.W. Visualization: A.C.E., M.W. Supervision: T.G.P.G., D.S., B.W.S., M.W. Project administration: T.G.P.G., E.B., J.R., D.S., B.W.S. Funding acquisition: W.B., E.B., J.R., T.G.P.G., D.S., B.W.S. The order of the first two authors' names was determined by a joint decision of both authors and M.W.

## Competing interests

The authors declare no conflict of interests.

## Additional information

[1]Department of Oncology and Children's Research Center, University Children's Hospital, University of Zurich, Zurich, Switzerland. [2]Division of Pediatric Hematology/Oncology, Department of Pediatrics, Inselspital, Bern University Hospital, Bern, Switzerland. [3]Hopp-Children's Cancer Center (KiTZ), Heidelberg, Germany. [4]Division of Translational Pediatric Sarcoma Research, German Cancer Research Center (DKFZ), German Cancer Consortium (DKTK), Heidelberg, Germany. [5]National Center for Tumor Diseases (NCT), NCT Heidelberg, a partnership between DKFZ and Heidelberg University Hospital, Heidelberg, Germany. [6]Medical Faculty, Ruprecht-Karls-University, Heidelberg, Germany. [7]Balgrist University Hospital, Faculty of Medicine, University of Zurich (UZH), Zurich, Switzerland. [8]Department of Medical Oncology and Hematology, University Hospital Zurich, Comprehensive Cancer Center Zurich, Zurich, Switzerland. [9]Department of Medical Oncology and Hematology, Cantonal Hospital St. Gallen, St. Gallen, Switzerland. [10]Division of Pediatric Hematology/Oncology, Children's Hospital of Eastern Switzerland, St. Gallen, Switzerland. [11]University Sarcoma Center Zürich (CCCZ), Balgrist University Hospital, University of Zurich, Zurich, Switzerland. [12]Swiss Center for Musculoskeletal Biobanking, Balgrist Campus AG, Zurich, Switzerland. [13]Department of Pathology and Molecular Pathology, University Hospital Zurich, Zurich, Switzerland. [14]Division of Biomedical Research, Medical University of Graz, Graz, Austria. [15]Institute of Pathology, Heidelberg University Hospital, Heidelberg, Germany. [16]Present address: Medical Oncology and Hematology, Cantonal Hospital Winterthur, Winterthur, Switzerland. [17]These authors contributed equally: Willemijn Breunis, Eva Brack. ✉e-mail: beat.schafer@kispi.uzh.ch; didier.surdez@balgrist.ch; marco.wachtel@kispi.uzh.ch

