## [Transparent Peer Review file · Nature Communications]

Patient-derived tumoroids from CIC::DUX4 rearranged sarcoma identify MCL1 as a therapeutic target

Corresponding Author: Dr Marco Wachtel

Version 0:

Reviewer comments:

Reviewer #1

(Remarks to the Author)

This is an interesting study that demonstrates the generation of organoids from sarcomas and identifies a novel therapeutic strategy. While the organoid generation data could be developed more the therapeutic portion and mechanistic investigation is intriguing and well done. Overall the manuscript could be presented better. The text could use headers to separate key findings. The figures are hard to read and many are too small. Please carefully proofread your manuscript.

What affected the success rate of organoid growth? Please give more details and provide a clear rational for decisions on protocols use.

How were the growth factors chosen for this study? Why not test others?

Why use viability when testing growth factors? Why not use proliferation and morphology as well? The authors should clarify which is affected when testing growth factors. How many passages were done with the growth factors? The FGF finding is intriguing but I am not convinced that it reduces viability. What would be a mechanistic reason for this finding?

Could the DNA/RNA/methylation data be used to optimize the media?

For synergy experiments please calculate using a few more methods such as ZIP, HSA, Loewe.

Is the DNA methylation primary data shown anywhere?

Fig 2E please label top 5 or 10 genes (both up and down regulated). Please comment on these findings. How does that correlate to the know biology of these tumors?

Fig 2F this data could be presented as a supplemental table. It is hard to read in the figure. Also minor issue but the dots for KEGG are orange and should be red to be consistent with the legend.

Please double check spelling and grammar. Also, triple check all numbers and results. For example a Pearson correlation is missing the "0."

Many figures are hard to read with small text and some legends are missing. Please consider reformatting as some of these will be hard to read in a publication format. Also some information shown in the figures are not discussed in the text such as the success rates in figure 1.

Reviewer #2

(Remarks to the Author)

Breunis and colleagues' manuscript investigates the application of CIC::DUX4 tumoroids in assessing potential sarcoma treatments. They particularly highlight MCL-1 inhibition as a promising therapeutic approach. The manuscript is well-written,

with clearly explained techniques. However a few experiments should be performed to improve its quality.

Major comments:

- In this paper the authors use tumoroids as models for therapy guidance but taking into account that on average they need 113 days to reach enough cells for drug testing, it is unlikely that sarcoma patients could benefit from this approach and should be corrected in the manuscript.
- The authors should provide evidence to demonstrate that CDS are more dependent on MCL1 than ES, and evaluate basal levels of MCL1 in both types of tumors.
- A comparison between MCL1 expression in the primary tumor and the tumoroid should be provided.
- Taking into account that the BCL2 family of proteins is a complex interactome, the authors should analyze the expression levels of the BCL2 family of proteins when performing CRISPR-Cas9 gene editing of MCL-1.
- The authors use microscope images or Cell Titer Glo to assess cell death, which is not acceptable. Other cell death measurements such as Annexin / PI should be provided in figure 5F and 6.

Minor comments:

- In Figure 5G-H quantification of MCL1 should be provided as the decrease in MCL1 is not clear.
- In Figure 6B a color legend should be provided.
- In Figure 6 the authors claim that the combination of chemotherapeutic agents with a MCL1 inhibitor provide no benefit. However, VIDE combination with MCL1 inhibition data is missing.

Reviewer #3

(Remarks to the Author)

Summary: This a very well written manuscript that highlights the power of tumoroid models to detect unique drug sensitivities in ES vs CDS. Importantly, they identify MCL1 as new drug target for CDS, an ultra-rare sarcoma subtype that is notoriously drug-resistant and associated with a poor prognosis. The authors rigorously pursue the role MCL1 plays in cell survival using genetic methods and drug testing. Overall, this manuscript nicely highlights the importance of re-classifying CDS as a distinct sarcoma subtype, apart from ES, with unique drug sensitivities.

A minor weakness includes the lack of comparison in drug sensitivity between patients and tumoroids, not unexpected given the rarity of CDS and dearth of investigational agents studied in the clinic for this disease.

Comments:

1. The abstract incorrectly states that metastatic ES lack systemic treatment options. Admittedly, survival is much worse for metastatic vs. localized ES because the former group of patients never achieve an NED status post chemo. That said, patients with metastatic ES are treated with the same regimens used for localized ES and benefit substantially.
2. I'd suggest refraining from the term "co-clinical" drug response profiling since this manuscript made no comparison between patient and tumoroid sensitivity. Co-clinical trials certainly may become the norm in the future but aren't yet validated.
3. Sarcoma tumoroids have been established on a large scale by another group at UCLA. The authors should discuss this work in the context of their own manuscript. What are the major differences in the establishment of the tumoroids? Are there differences in subtypes studied in either paper?
 - a. Al Shihabi, Ahmad, et al. "The landscape of drug sensitivity and resistance in sarcoma." *Cell stem cell* (2024).
4. I thought the investigation of growth factors effects was interesting. The authors may wish to comment on whether those factors would have affected drug sensitivity to agents targeting the PI3K/Akt/mTOR pathway.
5. Is there a molecular rationale for the downregulation of EWSR1::FL1 target genes due to bFGF? Also, it is surprising that IGF did not have an effect since IGF1R is strongly upregulated in EwS?
6. The schema in Figure 1 is unreadable. Perhaps it should be bigger. The fonts in figure 1 are also too small, especially 1c and 1d.
7. Might add a sentence explaining why the Archer FusionPlex analysis detected the CDS translocations while other technologies didn't.
8. In figure 2E, it would be a good check to call out some of the top genes distinguishing ES and CDS. Again, the font is quite small in Figure 2. Please also provide a table of genes comparing ES and CDS, as well as the genes used in the heatmap of figure 2.
9. How different are the drug list compared to the DepMap data? Is there common overlap between gene and drug targets? Are there differences in tumoroid response and cell line responses?
10. It would be clearer if the authors described how dDSS was calculated and the significance of high and low values of dDSS.
11. Is MCL1 a dependency for other cells in DepMap? It would be interesting to see how well it compares to the more recent PRISM from the Broad.
12. The knockdown of CIC::DUX4 to confirm MCL1 as a target was well executed.
13. Katia Scotlandi showed that CDS may be sensitive to PI3K/Akt/mTOR inhibitors. Has this been tested with the MCL1 antagonist in the tumoroid model? It would be great to include this combination in future studies.
14. Are there cells from the tumor microenvironment in the early passages of the tumoroids? Is there a known clonal

selection, detected by RNA-seq, as your group passages the tumoroids?

15. The font sizes for the x-axis for most figures are too small. Please review the fonts for all labels. They are not readable. I increased the figure to 200% before even being able to read it. Example: Figure 5C.

16. The second to the last sentence before the Discussion lists "Figure D". Is the figure number missing?

17. The clause on page 12, stating that suitable ex vivo culture conditions haven't been determined for 2D or 3D non-RMS models isn't entirely accurate. Numerous ES models exist in 2D, 3D, and as PDXs under various media conditions.

18. Page 14: While true, that "MCL1-directed agents with a "short half-life and rapid systemic clearance might circumvent the cardiac side effects," dose reductions are just as likely to reduce the on-target anti-MCL1 effect and limit drug efficacy.

Reviewer #4

(Remarks to the Author)

Reviewer #5

(Remarks to the Author)

Version 1:

Reviewer comments:

Reviewer #1

(Remarks to the Author)

The authors adequately addressed my questions and comments. Thank you.

Reviewer #2

(Remarks to the Author)

The authors addressed all our comments.

Reviewer #3

(Remarks to the Author)

The authors have significantly improved the manuscript and adequately responded to my concerns. Therefore, I recommend the manuscript for publication.

Reviewer #4

(Remarks to the Author)

Reviewer #5

(Remarks to the Author)

General comments

We exchanged the table with the sarcoma classifier scores in Figure 2A by a t-SNE plot integrating the data from the tumoroids with the ones from the sarcoma classifier reference cohort (n = 1,077). The original table is shown as Supplementary Table 2. We added a description of the procedure to the methods section on page 18.

Changes in the text are highlighted in red.

Replies to reviewer comments

Reviewer #1 (Remarks to the Author):

We would like to thank the reviewer very much for his work, which has contributed to a significant improvement of the manuscript.

This is an interesting study that demonstrates the generation of organoids from sarcomas and identifies a novel therapeutic strategy. While the organoid generation data could be developed more the therapeutic portion and mechanistic investigation is intriguing and well done. Overall the manuscript could be presented better. The text could use headers to separate key findings. The figures are hard to read and many are too small. Please carefully proofread your manuscript.

-> We have added headers to the results part of the text.

-> We have enlarged the figures to make them easier to read.

-> We also have proofread the manuscript and have corrected grammatical and spelling mistakes.

What affected the success rate of organoid growth? Please give more details and provide a clear rationale for decisions on protocols use.

-> That's indeed a very relevant question, however, the factors that influence the success rate of organoid growth are still not known in detail. In general, the amount and quality of the starting material, the number of live cells as well as cell-intrinsic (tumor aggressiveness) and extrinsic (tumor microenvironment) factors might be important aspects. We further discuss this topic in the discussion part of our manuscript, and newly added the cell-intrinsic and extrinsic aspects. The revised sentence on page 13 states: "While the exact reason for this is unclear in individual cases, the number and viability of cells isolated from biopsy samples as well as cell-intrinsic (tumor aggressiveness) or extrinsic (tumor microenvironment) factors may play an important role. Therefore, obtaining additional biopsies could further optimise the success rate and accelerate model generation."

-> The culture protocol we used is based on the organoid culture conditions originally developed in Hans Clevers' lab, first for normal epithelium and later also applied to different types of carcinoma. These conditions are now widely used as a basis for organoid culture systems across a broad range of cell types. In a previous study on the generation of ex vivo models of Rhabdomyosarcoma (PMID: 32934208), we systematically compared different culture conditions and found that (adapted) Clevers conditions are the most effective. Therefore, they also served as the basis for the study here.

In the original Clevers protocol, the cells were cultivated in a 3D Matrigel matrix in Advanced DMEM/F12 medium supplemented with the serum substitute B-27, N2 supplement, Nicotinamide, N-acetylcysteine, as well as various growth factors and signaling molecules including EGF, bFGF, Noggin (inhibitor of BMP signaling), Wnt3A and R-Spondin (activators of wnt signaling) and two small molecule inhibitors (TGFB inhibitor A83-01 and ROCK inhibitor Y-27632). Since there is no clear link between wnt and BMP signaling and cell proliferation in the case of EwS (no data is available for CDS), we omitted the corresponding signaling molecules and just tested bFGF, EGF and IGF1. Furthermore, since Y-27632 is thought to protect cells from anoikis immediately after dissociation, but is not required at later time points, we added this inhibitor only during the first 2-3 days after seeding to the medium.

To make this history clearer, we added the following sentences on page 6: "The basis were conditions that were originally optimized for epithelial and carcinoma organoids, which are now widely used to culture organoids derived from various tissues, including sarcomas^{22, 23, 24, 25, 26}." We also updated the methods part describing the organoid culture on page 17 with the following missing information: "To prevent anoikis after tissue dissociation, 10 μ M ROCK inhibitor Y-27632 was added to the medium for the first two to three days of culture."

How were the growth factors chosen for this study? Why not test others?

-> We tested bFGF and EGF, as they are widely used for organoid culture. We also tested IGF1 since the IGF1R pathway plays an important role in EwS as well as in other sarcoma entities. To clarify our selection, we added the following sentence on page 6: "We chose bFGF and EGF, both widely used in organoid protocols, and IGF1, as the IGF1R pathway plays a crucial role in EwS and other sarcoma entities."

We did not test additional growth factors, as the cells grew well under the applied conditions, and further optimization of the medium was not a priority.

As suggested by this reviewer below, we now also used our RNA-seq data to identify highly expressed growth factor receptors to guide the selection of additional growth factors (see also answer below). However, no receptor showed consistently high expression across models and no other growth factor was tested therefore.

Why use viability when testing growth factors? Why not use proliferation and morphology as well? The authors should clarify which is affected when testing growth factors. How many passages were done with the growth factors? The FGF finding is intriguing but I am not convinced that it reduces viability. What would be a mechanistic reason for this finding?

-> Initially, we aimed to establish conditions for propagating patient cells ex vivo and did not plan to characterize the effects of the growth factors in greater detail. However, as suggested by the reviewer, we have now conducted additional assays to characterize these effects. We focused on bFGF, which exerts the strongest influence on proliferation, and characterized its effect on one EwS (ES-ZH001) and one CDS (CDS-ZH001) model in detail. In particular, we measured proliferation and cell death by high content analysis and flow cytometry. Both assays revealed a strong induction of cell death in ES-ZH001 cells, whereas in CDS-ZH001 the primary effect is inhibition of proliferation. Taken together the data show that bFGF mediates its anti-proliferative effect by both induction of cell death as well as inhibition of proliferation, with the relative contribution varying between cell models. We have included these new data in the supplementary Figures S2a-d and added the following sentence to the text on page 6: "...in case of ES-ZH001 through rapid and strong induction of cell death as detected already after four days of treatment (Fig.1d and Supplementary Figure S2a-d), a finding consistent with published data on EwS cell lines^{28, 29}."

-> The initial growth factor test involved treating cells for 7 days without passaging as originally stated on page 6.

-> The anti-proliferative effect of bFGF on EwS has already been described for some EwS cell lines (PMID: 11085540, PMID: 2824563). We added the two references in the text to clarify this. In addition, our group has found a similar anti-proliferative effect of bFGF in some RMS models (PMID: 32934208). Together with the finding that some CDS models are also affected, this suggests that the anti-proliferative effect of bFGF might be common in sarcoma. The underlying mechanism is still unclear, but would certainly be worth further investigation. However, we believe that this is beyond the scope of the present study.

Could the DNA/RNA/methylation data be used to optimize the media?

-> Yes, we believe that this is an excellent suggestion. Hence, we used the RNA-seq data to analyze the expression of growth factor receptors (see table below). This analysis indeed revealed a correlation between the expression of growth factor receptors and the response to the corresponding growth factor in some cases. In particular, the ES-ZH001 tumoroid, which is most inhibited by bFGF, has the highest expression of FGFR1

among the EwS models, while the expression of other FGFRs is generally markedly lower. This would suggest that FGFR1 activation drives the anti-proliferative response to bFGF in EwS cells.

However, the analysis also revealed considerable heterogeneity in growth factor receptor expression across models. No single receptor was consistently expressed in all models. In addition, as the RNA level does not always accurately reflect the protein level, proteome or surfaceome data would be necessary to determine the expression levels of the various receptors more accurately, which was not available here. Further optimization of culture conditions based on these data would therefore require a model-specific approach and we decided against testing individual growth factor-model combinations.

Table Expression levels of growth factor receptors in CDS and EwS.

	ES-ZH001	ES-ZH002	ES-ZH004	ES-ZH007	ES-ZH008	ES-ZH009	CDS-ZH001	CDS-ZH002	CDS-ZH003	Patient CDS	Patient CDS
EGFR	300	729	7	28	4	1	2238	135	4956	894	280
ERBB2	3682	5353	6379	3117	3998	4852	2176	6271	1170	562	4936
ERBB3	9	56	143	616	8	77	101	175	77	197	467
ERBB4	8	48	18	189	197	94	0	2	4	4	69
PDGFRA	71	27	133	317	6	93	11359	3596	7033	2067	7271
PDGFRB	249	6691	2581	2744	182	2178	387	946	689	2691	4048
FGFR1	37358	12238	3760	5044	8408	8353	51704	38321	25889	9485	31330
FGFR2	0	2	168	13	4	485	1	1859	265	154	2381
FGFR3	12	579	436	864	1015	29	305	5866	5742	959	3644
FGFR4	865	914	586	659	184	204	339	855	452	51	245
FLT1	58	2	1	0	0	1	81	17	1258	2033	1852
KDR	2	37	29	15	0	127	66	25882	18	3119	4949
FLT4	0	1463	0	6	38	443	1299	8029	32150	4831	12698
INSR	3609	1578	5212	7595	5431	3672	6883	6304	5566	1468	4987
IGF1R	8031	7113	5802	3661	2219	2072	5638	8283	8859	1450	5096
MET	222	74	1713	1393	19	738	1	614	76	324	283
NTRK1	325	135	112	750	84	69	174	2	4	19	11
NTRK2	3	11	0	2	11	7	4	490	529	61	198
NTRK3	16	1	0	0	545	0	10	5	6	121	28
EPHA1	9	72	51	26	15	228	6	13	33	23	92
EPHA2	5600	4272	4168	11102	289	1631	5700	40594	8079	4235	15857
EPHA3	419	2563	4472	3671	931	1228	30895	317	1395	478	312
EPHA4	8768	3956	24451	8729	14524	6048	0	3627	3846	483	1222
EPHA5	133	1094	47	58	588	15	0	264	0	0	106
EPHA6	0	3	0	0	0	0	1	0	3	0	1
EPHA7	1080	297	41	102	4	734	0	3896	278	283	727
EPHA8	0	0	0	0	0	14	0	2	20	0	0
EPHA10	169	0	5	0	40	16	0	27	0	0	18
EPHB1	394	1	117	4	31	65	0	3404	210	76	817
EPHB2	3284	511	1366	1927	67	3948	198	5371	1431	539	1518
EPHB3	15272	441	17996	31413	1706	5703	13654	14695	9651	8467	6780
EPHB4	14169	11144	13302	16208	11986	8276	8279	11227	13383	5213	6813
EPHB6	27	43	138	21	125	22	0	0	1	29	57
RET	268	311	2537	7871	5	444	425	638	134	195	445
ALK	1681	146	630	3710	926	93	1	752	34	18	634
DDR1	2089	2048	3939	3113	4713	1210	270	1919	625	216	1862
DDR2	54256	6660	53142	28976	8684	9240	51345	8630	36154	4944	5360
TGFBR1	1686	359	278	190	327	3837	2072	4532	2458	2177	1999
TGFBR2	110	392	46	23	11	275	5985	20107	16946	6638	9295
TNFRSF1A	3520	4544	4136	4949	10658	2673	16705	11282	20134	8205	8392
TNFRSF1B	26	32	0	26	2	3	548	395	1475	1448	11425
IL2RA	2	0	1	0	0	0	0	0	0	222	865
IL4R	257	52	516	188	150	314	771	782	2007	762	1989
IL6R	0	32	11	501	2820	49	109	198	216	469	2108
ITGAV	5302	2426	6514	4116	2496	2266	2686	8842	8873	3655	7253
ITGB3	6	9	2	379	3	1	83	1307	28	149	478
LPAR1	2837	5453	6770	5266	3111	934	431	5059	1086	700	1668
LPAR2	1519	1238	1925	1206	1275	1877	92	1379	186	333	1155
LPAR3	0	47	0	3	163	21	3	3	173	196	26
LPAR4	407	169	86	308	396	48	0	0	0	12	2
LPAR5	12	8	30	196	62	27	3	2	5	83	67
LPAR6	0	726	1174	155	1310	81	536	129	204	705	816
S1PR1	3	26	151	36	125	3	5409	10	882	1860	803
S1PR2	600	419	631	1454	457	1138	1583	750	1726	943	1253
S1PR3	6479	412	509	1536	4914	487	7615	1887	9282	2056	1582
S1PR4	0	0	0	2	0	1	3	0	0	97	108
S1PR5	306	555	249	1033	1180	296	1270	291	475	506	392
EPOR	2106	177	305	487	6371	417	238	254	264	138	255
CSF3R	0	0	6	117	12	130	61	25	17	167	2056
MPL	26	5	27	4	36	0	8	11	10	9	17
ESR1	2	4	8	14	20	1	3	13	4	25	33
PGR	0	0	0	0	0	0	0	30	0	3	6
AR	3	3	1361	1738	1364	9	0	74	17	22	34

For synergy experiments please calculate using a few more methods such as ZIP, HSA, Loewe.

-> In addition to the Bliss scores, we now have also calculated the ZIP and Loewe scores for all synergy tests and included these data as Supplementary Figure S11A-B in the manuscript. The new figure is described in the paragraph on page 12: "Bliss, ZIP, and Loewe synergy score analyses showed no synergy between S64315 and chemotherapeutic agents. In contrast, the combination of S64315 with venetoclax demonstrated clear synergism in two of the four CDS models, while its combination with PQR514 showed synergism in one model (Fig.6b-c and Supplementary Fig. S11a-b)."

Is the DNA methylation primary data shown anywhere?

-> Yes, the methylation data will be available in the EGA database under the accession number EGAS00001008039. We added the number to the data accessibility section.

Fig 2E please label top 5 or 10 genes (both up and down regulated). Please comment on these findings. How does that correlate to the know biology of these tumors?

-> We have now labeled the top 10 up- and downregulated genes in Figure 2e. While we believe that drawing major conclusions from this small number of genes is challenging, we agree with the reviewer that analyzing differentially expressed genes between EwS and CDS could provide valuable insights into the biology of these tumors. This aspect is already addressed by the GSEA approach (old Figure 2f, now Supplementary Table 5), which identified pathways associated with a larger set of differentially expressed genes. These findings are discussed in the text on pages 7 and 8.

Fig 2F this data could be presented as a supplemental table. It is hard to read in the figure. Also minor issue but the dots for KEGG are orange and should be red to be consistent with the legend.

-> As suggested by the reviewer we have moved the GSEA data to the Supplementary Table 5.

Please double check spelling and grammar. Also, triple check all numbers and results. For example a Pearson correlation is missing the "0."

-> We thank the reviewer for carefully reading our manuscript. We now double checked and corrected spelling and grammar of our text, as well as numbers and results and added the missing 0.

Many figures are hard to read with small text and some legends are missing. Please consider reformatting as some of these will be hard to read in a publication format. Also some information shown in the figures are not discussed in the text such as the success rates in figure 1.

-> We increased the size of the figures and completed all the legends.

-> The success rate shown in Figure 1 is described in the results section. We now added the percentages on page 6 to make it more clear: "In seven EwS cases (50%) and all four CDS cases (100%) we could successfully establish a tumoroid model (Fig.1b and c)." We refined the discussion of the success rate in the discussion section on page 13:" While the exact reason for this is unclear in individual cases, the number and viability of cells isolated from biopsy samples as well as cell-intrinsic (tumor aggressiveness) or extrinsic (tumor microenvironment) factors may play an important role. Therefore, obtaining additional biopsies could further optimise the success rate and accelerate model generation."

Reviewer #2 (Remarks to the Author):

We would like to thank the reviewer very much for his work, which has contributed to a significant improvement of the manuscript.

Breunis and colleagues' manuscript investigates the application of CIC::DUX4 tumoroids in assessing potential sarcoma treatments. They particularly highlight MCL-1 inhibition as a promising therapeutic approach. The

manuscript is well-written, with clearly explained techniques. However a few experiments should be performed to improve its quality.

Major comments:

- In this paper the authors use tumoroids as models for therapy guidance but taking into account that on average they need 113 days to reach enough cells for drug testing, it is unlikely that sarcoma patients could benefit from this approach and should be corrected in the manuscript.

-> We agree with the reviewer that the average development time for the ex vivo model described here is not fast enough for therapy guidance. However, one major aim of our study was to test our entire drug library with around 250 drugs in order to characterize the tumoroid models in as much detail as possible and not to be able to provide data as quickly as possible.

Based on data from the leukemia group at the Children's Hospital Zurich as well as our own data, we will start a clinical study at our institution this summer to test the performance of the drug profiling platform in a co-clinical setting in solid and blood cancers (KidScan study). In the context of this study, the aim is to be able to deliver data within 4 weeks. In case of tumor resections, the number of cells is often sufficient to test a larger number of drugs in a direct approach (see Figures 1a and 3a and also a recent publication from a group at UCLA (PMID: 39305899). In case of biopsies, on the other hand, the number of cells is certainly limiting, so that in a direct approach only a small number of drugs can be tested in each case. We hope that our data will stimulate surgeons to take more biopsies for functional characterization, as is already the case today for molecular characterization (RNA-seq, WES and methylation profiling). We believe that the present work is a basis for all these approaches.

To make these aspects more clear, we re-formulated the discussion on page 13: "...At this stage, the number of available cells is sufficient to test a limited set of drugs, such as the chemotherapeutics used as first-line therapy. This timeframe may also be compatible with a co-clinical approach. Here, however, we expanded the cells for an additional one to two passages to obtain at least 10 million cells, a number required to test our drug library of 245 drugs. The average duration for this approach of 130 days is too long for a co-clinical application for therapy guidance in individual patients. However, this approach allowed us not only to test clinically approved compounds, but also to characterize the models in great detail and gain insights into unidentified relevant pathways."

- The authors should provide evidence to demonstrate that CDS are more dependent on MCL1 than ES, and evaluate basal levels of MCL1 in both types of tumors.

-> While IC50 values are generally lower in CDS than EwS, some of the latter are also relatively sensitive to MCL1 inhibition when compared to other tumors. Accordingly, EwS is enriched as a sensitive entity to S63845 in DepMap. Therefore, we believe that comparing CDS tumors to a broader range of tumor types provides a more representative classification of their sensitivity to MCL1 inhibitors. To do this, we leveraged the new MCL1 inhibitor BRD-810, which is structurally unrelated to S63845/S64315 and for which IC50 data from 696 cell lines is available (PMID: 39179926). We treated 4 CDS and 3 EwS models with BRD-810 and compared the IC50 values with those of the 696 cell lines. Strikingly, the IC50s of 3 out of 4 CDS models were among the most sensitive cells when compared to these 696 cell lines (7th, 16th and 32nd in the ranked list), while the tested EwS models had much higher IC50s under these conditions. This novel data is shown in the new supplementary Figures S5a-c. To describe the data, we added the following sentence on page 9: "Tests with BRD-810, a structurally unrelated MCL1 inhibitor, for which PRISM data from 696 cell lines are available³⁸, further confirmed these findings. For CDS-ZH001, CDS-ZH003, and an additional CDS model (MUG CIDUS; established at a different center), IC50 values of BRD-810 ranked 7th, 13th, and 32nd when added to the list of 696 tumor cell lines (Supplementary Figure S5a-c)."

-> Basal levels of MCL1 are shown in Figure 5a (protein) and 5b (RNA) as well as in Supplementary Figure S7b (RNA in comparison to a large cohort of normal tissue and tumors).

- A comparison between MCL1 expression in the primary tumor and the tumoroid should be provided.

-> We added the normalized read counts from the two available primary tumors to Figure 5b.

- Taking into account that the BCL2 family of proteins is a complex interactome, the authors should analyze the expression levels of the BCL2 family of proteins when performing CRISPR-Cas9 gene editing of MCL-1.

-> As requested by the reviewer, we analyzed protein levels of additional BCL2 family proteins after knockout of MCL-1 in CDS-ZH003 cells. This analysis revealed no effect on BCL2 and BCL-XL levels. We concluded that MCL1 depletion does not lead to major changes in the expression levels of the other relevant BCL2 family members. We show these data in the novel supplementary Figure S3b and added the following text to the manuscript on page 10: "while expression levels of BCL2 and BCL-XL were not significantly affected under these conditions (Supplementary Figure S6)."

- The authors use microscope images or Cell Titer Glo to assess cell death, which is not acceptable. Other cell death measurements such as Annexin / PI should be provided in figure 5F and 6.

-> As requested by the reviewer, we now have assessed the number of live and dead cells in these experiments. For Figure 5f, we stained the cells with Hoechst and PI and used image-based high-content analysis to quantify live and dead cells. We replaced the microscope images with this novel data in the new Figure 5e and moved the images to the new Supplementary Figure S8a. In addition, we also performed Annexin V/PI stainings and quantified live and dead cells by flow cytometry. These data are shown in the novel Supplementary Figures S8b-c. The data from the two new assays are in good agreement and show that shDUX4-1 and -3 both induce cell death, whereas shDUX4-2 does not, which is in agreement with the silencing efficiency.

For Figure 6, we stained the cells with Hoechst and PI and used high-content analysis to assess the effects of the drug combinations on the different tumor models. We replaced the cell viability data with the new data in both Figures 6b and c. This new data is similar to the viability data shown before.

Minor comments:

- In Figure 5G-H quantification of MCL1 should be provided as the decrease in MCL1 is not clear.

-> We have added the quantification of MCL1 in Figure 5h. Since the old blots were no longer available and the cell lysates were quite old, we repeated the experiment and replaced the blots in Figure 5f.

- In Figure 6B a color legend should be provided.

-> We added the color legend to Figure 6b.

- In Figure 6 the authors claim that the combination of chemotherapeutic agents with a MCL1 inhibitor provide no benefit. However, VIDE combination with MCL1 inhibition data is missing.

-> The reviewer is correct, we did not combine the MCL1 inhibitor with all four chemotherapeutics at once, as such a combination would not allow for the calculation of a synergy score. However, we combined the MCL1 inhibitor with each chemotherapeutic agent individually and calculated the corresponding synergy scores. This data is shown in Figure 6c.

Reviewer #3 (Remarks to the Author):

We would like to thank the reviewer very much for his work, which has contributed to a significant improvement of the manuscript.

Summary: This a very well written manuscript that highlights the power of tumoroid models to detect unique drug sensitivities in ES vs CDS. Importantly, they identify MCL1 as new drug target for CDS, an ultra-rare

sarcoma subtype that is notoriously drug-resistant and associated with a poor prognosis. The authors rigorously pursue the role MCL1 plays in cell survival using genetic methods and drug testing. Overall, this manuscript nicely highlights the importance of re-classifying CDS as a distinct sarcoma subtype, apart from ES, with unique drug sensitivities.

A minor weakness includes the lack of comparison in drug sensitivity between patients and tumoroids, not unexpected given the rarity of CDS and dearth of investigational agents studied in the clinic for this disease.

-> We completely agree with the reviewer that such a comparison would be of great interest; however, a clinical application was not possible here.

Comments:

1. The abstract incorrectly states that metastatic ES lack systemic treatment options. Admittedly, survival is much worse for metastatic vs. localized ES because the former group of patients never achieve an NED status post chemo. That said, patients with metastatic ES are treated with the same regimens used for localized ES and benefit substantially.

-> The reviewer is correct that we somewhat understated the therapeutic options for these patients in the abstract. We wanted to emphasize that only a few high-risk patients can be cured with these treatments. We have now revised the abstract and mainly highlight that the prognosis for these patients remains poor despite the available therapeutic options. The sentence now states: "High-risk sarcomas, such as metastatic and relapsed Ewing and CIC-rearranged sarcoma, still have a poor prognosis despite intensive therapeutic regimens."

2. I'd suggest refraining from the term "co-clinical" drug response profiling since this manuscript made no comparison between patient and tumoroid sensitivity. Co-clinical trials certainly may become the norm in the future but aren't yet validated.

-> We agree with the reviewer that in our study no co-clinical trial was conducted and therefore, as suggested by the reviewer, we removed the term "co-clinical" in all cases where a parallel treatment of patients could be meant. In the coming months a new study named KidsCan will start at our institute. This study has the aim to test the performance of ex vivo drug profiling in a co-clinical setting in pediatric solid and blood cancers.

3. Sarcoma tumoroids have been established on a large scale by another group at UCLA. The authors should discuss this work in the context of their own manuscript. What are the major differences in the establishment of the tumoroids? Are there differences in subtypes studied in either paper?

a. Al Shihabi, Ahmad, et al. "The landscape of drug sensitivity and resistance in sarcoma." Cell stem cell (2024).

The point raised by the reviewer addresses an important aspect in the field of co-clinical DRP, namely the number of available cells. Since, in most cases, we obtained biopsies from patients, we aimed to establish stable ex vivo models that can be propagated over many passages (we have not yet experienced loss of any of our models due to a crisis event) and allow the cells to be expanded to larger numbers.

In contrast, Al Shihabi and colleagues cultured the cells for only 5 days (3 days in absence of treatment plus 2 days under treatment). This approach relates to the direct profiling approach mentioned in Figure 1a of our manuscript (lower arm in that figure) and necessitates enough cells from the patient tumor (threshold defined as 250`000 in the Al Shihabi paper). This is likely why in 183/194 (94%) cases in the Al Shihabi study surgical resections were used and only in 11 cases biopsies. Furthermore, it's not clear in how many of these 11 biopsy cases a drug profiling was ultimately possible, since, in 21 cases, there was an insufficient cell number for further analysis. Overall, material quantity is for both studies a limiting factor and a perspective for both studies could be to scale down well size (e.g.1536 well plates) or reduce number of tested drugs.

We added the following brief discussion of this topic to the discussion section on page 13, referring to the Al Shihabi paper as an example for the direct profiling approach with sarcoma: "The short-term approach, originally developed for blood cancer, is now increasingly used for different types of solid tumors, including sarcoma ²³. This approach requires sufficient cells from the patient tumor and is mainly suitable for tumor resections ²³. In

the case of biopsies, the number of cells is a limiting factor and ex vivo expansion is often necessary, which requires appropriate culture conditions ^{24, 25, 26}."

4. I thought the investigation of growth factors effects was interesting. The authors may wish to comment on whether those factors would have affected drug sensitivity to agents targeting the PI3K/Akt/mTOR pathway.

-> We pursued this idea and tested the influence of EGF and IGF on the cytotoxic effects of the pan-PI3K inhibitor PQR514 and the mTOR inhibitor everolimus in CDS-ZH003 cells. After treatment, we stained the cells with Hoechst and PI and used an image-based readout to determine the number of live and dead cells. This experiment showed no significant change in the dose-response curves, suggesting that under the applied conditions EGF and IGF have no major influence on sensitivity towards drugs targeting the PI3K/Akt/mTOR pathway (see Figure A below).

Figure A. Dose response curves of PQR514 (left panel) and everolimus (right panel) generated with the CDS model CDS-ZH003. Cells were treated in the presence or absence of EGF and IGF.

5. Is there a molecular rationale for the downregulation of EWSR1::FL1 target genes due to bFGF? Also, it is surprising that IGF did not have an effect since IGF1R is strongly upregulated in EwS?

-> We performed a Western blot to evaluate the effect of bFGF on EWS::FLI1 protein levels in ES-ZH001 cells (see Figure B below). This analysis revealed a bFGF-induced downregulation of EWS::FLI1 in these cells, which may explain the observed changes in target gene expression. We believe that the unexplored effects of bFGF on EWS::FLI1 protein levels represent an interesting topic, but also think that this is beyond the scope of the manuscript here. We therefore decided not to include these data in the manuscript.

Figure B. Western blot analysis of indicated proteins in cell lysate from ES-ZH001 cells after treatment with bFGF for indicated time periods.

We agree with the reviewer that the lack of response to IGF1 is unexpected. The known prominent role of IGF1R in Ewing sarcoma was the reason for us to test its effect on tumoroid growth. At this stage, we can only speculate on the underlying reasons for this observation. Potential explanations include autocrine pathway activation or the presence of IGF1 in Matrigel. Notably, clinical trials with IGF1R inhibitors (PMID: 22025149 and PMID: 37306107) have also shown highly heterogeneous responses in EwS. This underscores the need for further investigation into the dependency of EwS on this pathway.

6. The schema in Figure 1 is unreadable. Perhaps it should be bigger. The fonts in figure 1 are also too small, especially 1c and 1d.

-> We apologize for the suboptimal figure quality. We increased the overall size of the scheme in Figure 1 and also increased the font size. Additionally, we increased the font size in all the other panels of Figure 1.

7. Might add a sentence explaining why the Archer FusionPlex analysis detected the CDS translocations while other technologies didn't.

-> The fact that some common fusion callers fail to detect the CIC::DUX4 translocation has been described in a recent paper (PMID: 24950227). One possible reason for the poor performance of the fusion callers might be the complexity of the two *DUX4* loci, which contain numerous pseudogenes and are located on two chromosomes. We included the reference mentioned above and added the following sentence to the text on page 7: "... similar to another case described in the literature ³², possibly due to the complex structure of the *DUX4* loci, which contain numerous pseudogenes."

8. In figure 2e, it would be a good check to call out some of the top genes distinguishing ES and CDS. Again, the font is quite small in Figure 2. Please also provide a table of genes comparing ES and CDS, as well as the genes used in the heatmap of figure 2.

-> We now labeled the top 10 up- and downregulated genes (based on log fold change) in Figure 2e. In addition, we reorganize the entire figure and increased the font size in all panels. Furthermore, we replaced the table in figure 1a with a t-SNE plot integrating the methylation data from the tumoroids alongside methylation profiles

from the sarcoma classifier reference cohort (n = 1,077). We moved the original table with the classifier scores to the supplementary part (new Supplementary Table 2).

Finally, we added tables listing the genes from the heatmap and those that distinguish CDS from EwS (Supplementary Tables 3 and 4).

9. How different are the drug list compared to the DepMap data? Is there common overlap between gene and drug targets? Are there differences in tumoroid response and cell line responses?

-> As suggested by the reviewer, we compared our data to DepMap data. We used the Sanger dataset (GDSC1 and GDSC2) for this purpose, as it contains a large number of EwS models, unlike the PRISM dataset. Because only one CDS cell line was available in the GDSC1 dataset, and none in the GDSC2 dataset, we focused our analysis on EwS.

For EwS, we found 45 drugs in our library for which data was also available in the Sanger dataset. When including also other tumor types, the number of drugs available for all tumors dropped to 31. We used these 31 drugs and performed a Pearson correlation analysis to compare the mean IC50 values from our EwS tumoroids with those of the Sanger datasets. This analysis revealed that the Pearson correlation coefficient for the Sanger EwS dataset was among the highest. However, the correlations with the B-ALL and neuroblastoma datasets resulted in similarly high correlation coefficients (see Figure C below). Correlation coefficients were lower for other sarcoma entities as well as NSCLC and breast cancer. On the one hand, this demonstrates that drug response in tumoroids and cell lines is similar. However, because the number of available drugs for this analysis is small and many of these drugs are broadly cytotoxic chemotherapeutics, the response pattern lacks specificity and does not allow for tumor classification. We therefore believe that this comparison provides limited insight, and we have refrained from adding it to the manuscript.

Figure C. Pearson correlation coefficients for the correlation of average IC50 values from our own EwS and tumors from the GDSC dataset (Sanger).

10. It would be clearer if the authors described how dDSS was calculated and the significance of high and low values of dDSS.

-> The reviewer is right, the dDSS approach was not described in sufficient detail. We added the following text for clarification on page 8: "The drug sensitivity score (DSS) is computed as the normalized area under the curve (AUC) over the measured concentration range, scaling between 0 (minimal drug activity) and 100 (maximal drug activity). The dDSS score quantifies the difference between the drug response of a given sample of interest and the average drug response of a reference cohort 35. It is calculated by subtracting the average DSS of the reference cohort from the DSS of the sample of interest. Thus, the larger the difference to the reference cohort, the more exceptional the drug response of a given sample."

11. Is MCL1 a dependency for other cells in DepMap? It would be interesting to see how well it compares to the more recent PRISM from the Broad.

-> We agree with the reviewer that these are interesting aspects. According to the Chronos knockout data in DepMap, 300/1103 (27%) cell lines show a gene effect of < -1, demonstrating that also other cell lines depend on MCL1 expression. Unfortunately, there is no data on CDS cell lines available in DepMap nor in the literature; therefore, a direct comparison is not possible.

Data from the MCL1 inhibitor S63845 is indeed available in the most recent PRISM dataset; however, this dataset unfortunately also does not include CDS cells and the drug was only tested at 2.5 μ M. Therefore, no relevant comparison is possible with this dataset.

12. The knockdown of CIC::DUX4 to confirm MCL1 as a target was well executed.

-> We thank the reviewer for this comment.

13. Katia Scotlandi showed that CDS may be sensitive to PI3K/Akt/mTOR inhibitors. Has this been tested with the MCL1 antagonist in the tumoroid model? It would be great to include this combination in future studies.

-> We appreciate this valuable suggestion. In the combination screen shown in Figure 6A, PQR514 (PI3K inhibitor) was indeed among the drugs with the highest combinatorial effect when combined with the MCL1 inhibitor S64315, whereas BEZ235 (tested by the Scotlandi group) was not among the top hits. We highlighted PQR514 in Figure 6A and added the following sentence to the figure description on page 12: "Additionally, we identified several BET bromodomain inhibitors and the PI3K inhibitor PQR514 among the top hits."

Furthermore, we validated this finding with a combination matrix of S64315 and PQR514. This experiment demonstrated that, in some models, this combination was also synergistic, though to a lesser extent than S64315 and venetoclax, supporting the findings from the combination screen. We added the synergy scores for this combination in Figure 6C and updated the text on page 12 accordingly: "In contrast, the combination of S64315 with venetoclax demonstrated clear synergism in two of the four CDS models, while the combination with PQR514 showed synergism in one model (Fig.6B-C and Supplementary Figure S11A-B)."

We also included this aspect in the discussion on page 14: "We did not observe exceptional sensitivity to dactolisib either as single agent or in combination with S64315 in our CDS models; however, we detected a synergy between the PI3K inhibitor PQR514 and S64315 in one of the models."

14. Are there cells from the tumor microenvironment in the early passages of the tumoroids? Is there a known clonal selection, detected by RNA-seq, as your group passages the tumoroids?

-> We did not specifically study the stroma content in our cultures. However, while at the beginning of the ex vivo culture, stromal cells might be present, tumor cells outcompete these cells during prolonged culture. At later time points, the cultures contain almost exclusively tumor cells.

-> We also did not assess clonal selection during extended culture periods. We performed the molecular analysis only at one single time point.

15. The font sizes for the x-axis for most figures are too small. Please review the fonts for all labels. They are not readable. I increased the figure to 200% before even being able to read it. Example: Figure 5C.

-> We increased the font size of the x-axes in the figures wherever applicable, including Figure 5C (now Supplementary Figure S7).

16. The second to the last sentence before the Discussion lists "Figure D". Is the figure number missing?

-> Yes, the Figure number was indeed missing. We added the number Figure 6D.

17. The clause on page 12, stating that suitable ex vivo culture conditions haven't been determined for 2D or 3D non-RMS models isn't entirely accurate. Numerous ES models exist in 2D, 3D, and as PDXs under various media conditions.

-> The reviewer is indeed right. Hence, we have removed the sentence. We now state on page 13: "In the case of biopsies, the number of cells is a limiting factor and ex vivo expansion is often necessary, which requires appropriate culture conditions".

18. Page 14: While true, that "MCL1-directed agents with a "short half-life and rapid systemic clearance might circumvent the cardiac side effects," dose reductions are just as likely to reduce the on-target anti-MCL1 effect and limit drug efficacy.

-> In a recent paper (PMID: 39179926) from the group of Todd Golub, to which the above sentence refers, the authors demonstrate that their novel MCL1 inhibitor BRD-810 effectively kills tumor cells, while sparing cardiomyocytes due to its short half-life. In their study, dogs were repeatedly infused with the MCL1 inhibitor over 4 weeks at a concentration shown to effectively kill tumor cells in mouse models. Importantly, no cardiotoxic effects were observed.

Based on these data, the authors conclude that their novel inhibitor may have a therapeutic window, unlike other MCL1 inhibitors. We tested the effect of BRD-810 on our CDS models and could confirm the findings made with S63845/S64315 (Supplementary Figures S5A-C). These data might open a new clinical perspective with less toxic next generation MCL inhibitors.

We have now added the missing reference to the text.

Reviewer #4 (Remarks to the Author):

-> We appreciate the work of the reviewer that helped to improve our manuscript considerably.

Reviewer #5 (Remarks to the Author):

-> We appreciate the work of the reviewer that helped to improve our manuscript considerably.